

Freshening of the Labrador Sea as a trigger for Little Ice Age development
Montserrat Alonso-Garcia[1,2,3], Helga F. Kleiven[4], Jerry F. McManus[5], Paola Moffa-Sanchez[6],
Wallace Broecker[5] and Benjamin P.Flower[1]
1 College of Marine Science, University of South Florida, St. Petersburg, FL, US.
2 Instituto Português do Mar e da Atmosfera (IPMA), Div. de Geologia e Georecursos Marinhos,
Lisboa, Portugal.
3 Centro de Ciencias do Mar (CCMAR), Universidade do Algarve, Faro, Portugal
4 University of Bergen and Bjerknes Centre for Climate Research, Postboks 7803, 5020 Bergen,
Norway.
5 Department of Earth and Environmental Sciences, Lamont-Doherty Earth Observatory of
Columbia University, Palisades, NY  10964-8000 USA.
6 Department of Marine and Coastal Sciences, Rutgers University, New Brunswick, NJ, 08901,
USA.
*Correspondence to*: Montserrat Alonso-Garcia (montserrat.alonso@ipma.pt)
Abstract
Arctic freshwater discharges to the Labrador Sea from melting glaciers and sea-ice can have a deep
impact on ocean circulation dynamics in the North Atlantic modifying climate and deep water
formation in this region. In this study, we present for the first time a high resolution record of ice-
rafting in the Labrador Sea over the last millennium to assess the effects of freshwater discharges in
this region on ocean circulation and climate. The occurrence of ice-rafted debris (IRD) in the
Labrador Sea was studied using sediments from Site GS06-144-03 (57.29° N, 48.37° W, 3432 m
water depth). IRD from the fraction 63-150 µm show higher concentration during the intervals:
~1000-1100, ~1150-1250, ~1400-1450, ~1650-1700 and ~1750-1800 yr AD. The first two intervals
occurred during the Medieval Climate Anomaly (MCA), whereas the others took place within the
Little Ice Age (LIA). Mineralogical identification indicates that the main IRD source during the
MCA was SE Greenland. In contrast, the concentration and relative abundance of hematite-stained
grains (HSG) reflects an increase in the contribution of Arctic ice during the LIA.



The comparison of our Labrador Sea IRD records with other climate proxies from the subpolar North Atlantic allowed us to propose a sequence of processes that led to the cooling events during the LIA, particularly in the Northern Hemisphere. This study reveals that the warm climate of the MCA may have enhanced iceberg calving along the SE Greenland coast and, as a result, freshened the subpolar gyre (SPG). Consequently, SPG circulation switched to a weaker mode through internal feedbacks that reduced convection in the Labrador Sea decreasing its contribution to the Atlantic Meridional overturning circulation and, thus, the amount of heat transported to high latitudes. This mechanism very likely preconditioned the North Atlantic inducing a state in which external forcings (e.g. solar irradiance and volcanic input) could easily drive periods of severe cold conditions in Europe and the North Atlantic like the LIA. The outcomes of this work indicate that a freshening of the SPG may play a crucial role in the development of cold events during the Holocene, which may be of key importance for predictions about future climate.

Key words: Little Ice Age, Medieval Climate anomaly, Labrador Sea, ice-rafting

1. Introduction

The last millennium is a primary target in paleoclimate studies since this interval allows us to reconstruct the climate variability of our recent history and its impact on the development of our society. Moreover, climatic reconstructions of the last millennium combined with instrumental records constitute a remarkable framework to obtain a comprehensive understanding of the mechanisms that drive the Earth's climate and improve future climate predictions. The climate of the last millennium is characterized by a warm period called the Medieval Climate Anomaly (MCA) or Medieval Warm Period (~800-1200 yr AD), a cold interval called the Little Ice Age (LIA, ~1350-1850 yr AD) and the 20[th] century warming trend (e.g. Mann et al., 2009; Wanner et al., 2011). According to historical records, these climate oscillations affected human development in Europe, in particular, the Norse expansion in the North Atlantic (Ogilvie et al., 2000). The warm conditions of the MCA promoted the colonization of Iceland and Greenland by the Norse and the exploration of North America during the 9th to 12th centuries, whereas the deterioration of climatic conditions at the beginning of the LIA forced them to abandon the Greenland settlements by the end of the 15th century (Kuijpers et al., 2014; Ogilvie et al., 2000).

Reconstructions of ocean and land temperature show the LIA cooling was neither spatially nor temporally uniform (Bradley et al., 2003; PAGES 2k Consortium, 2013; Wanner et al., 2015;



Wanner et al., 2011) and, therefore, there is an open debate on the forcings that may have triggered
these climate oscillations. Reduced solar irradiance and the occurrence of explosive volcanic
eruptions are the two most commonly examined forcings (e.g. Bond et al., 2001; Miller et al., 2012)
due to the impact they may have on atmospheric dynamics. Other forcings such as the internal
dynamics of the oceanic and atmospheric systems (such as the North Atlantic Oscillation-NAO-,
Arctic Oscillation-AO-, Atlantic Multidecadal Oscillation-AMO-, El Niño-Southern Oscillation-
ENSO-, or the monsoonal regimes) have also been considered to play a major role driving climate
oscillations during the last century (see review in Wanner et al., 2011). Freshwater discharges to the
North Atlantic may also be the drivers of climate change through their impact on sea surface
circulation and deep water convection, which in turn may slowdown the Atlantic Meridional
Overturning Circulation (AMOC) (Manabe and Stouffer, 1995). Particularly, the Labrador Sea is
very sensitive to increases in freshwater and sea ice input. Deep water formation in the Labrador
Sea contributes 30% of the volume transport of the deep limb of the AMOC (Rhein et al., 2002;
Talley, 2003), and freshwater input to this region reduces deep convection and deep water
formation, slowing down the overturning circulation and oceanic heat transport by up to 27 and 15
%, respectively (Born et al., 2010). The decrease in heat export from low to high latitudes modifies
regional climate by cooling the western North Atlantic which, in turn, influences the climate of the
whole North Atlantic (Born et al., 2010). A recent example of this phenomenon may be the Great
Salinity Anomaly (Dickson et al., 1988). During this event, vast amounts of Arctic sea ice and
freshwater were delivered to the Labrador Sea, mainly via the East Greenland Current (EGC),
freshening the subpolar gyre (SPG) and decreasing winter convection and deep water production. A
recent study of the last 50 years also shows a close relationship between fresh water fluxes from the
Arctic and reductions in deep water formation in the Labrador Sea (Yang et al., 2016).
Recently, a lot of attention has been paid to the dynamics of the SPG and its relationship with
climate (e.g. Born and Stocker, 2014). Instrumental records and modern observations show a close
link between decadal climate variability and SPG dynamics (e.g. Hakkinen and Rhines, 2004;
Sarafanov, 2009), and rapid climate change reconstructions throughout the last glacial cycle have
been interpreted as a consequence of changes in the SPG dynamics (Moffa-Sanchez et al., 2014a;
Mokeddem and McManus, 2016; Mokeddem et al., 2014; Moros et al., 2012; Thornalley et al.,
2009). Variations in the strength and shape of the SPG may also impact deep convection in the
Labrador Sea, therefore influencing deep water production and AMOC (Böning et al., 2006; Hatun
et al., 2005), which eventually affects climate through the reduction of heat transported from low to
high latitudes. A shift to weak SPG circulation has been inferred using deep-sea corals between the
MCA and the LIA (Copard et al., 2012) and model simulations suggested this weakening of the



SPG was the main driver of the LIA due to the decrease in meridional heat transport to the subpolar North Atlantic (Moreno-Chamarro et al., 2016). Moreover, the occurrence of unusually cold winters in Europe during the last 100 years has been associated with atmospheric blocking events in the North Atlantic, which are high pressure systems that alter the normal westerly wind circulation in this region (Häkkinen et al., 2011). These events are associated with negative NAO, may modify surface circulation in the North Atlantic and are linked to cold winter temperature in western Europe (Shabbar et al., 2001). Periods of intense and persistent atmospheric blocking events very likely developed during the LIA due to the influence of low solar irradiance and weak SPG circulation, causing decadal intervals of severe cooling in Europe (Moffa-Sanchez et al., 2014a).

In this work we used a sediment core from the Eirik Drift, in the Labrador Sea, to reconstruct the ice-rafting occurrence during the last 1200 yr and examine its impact on SPG dynamics and climate. The presence of ice-rafted debris (IRD) is a proxy for iceberg and sea ice discharges. Our IRD record from the Eirik Drift indicates ice export to the Labrador Sea and allows us to infer periods of enhanced freshwater discharges. Previous Holocene multi-proxy records (including IRD records) from the North Atlantic pointed to the linkage between cooling events and low solar irradiance values (Bond et al., 2001). However, this hypothesis has been challenged by the fact that ice-rafting reconstructions in the Northern North Atlantic show different trends between the eastern and western regions during the Holocene (Moros et al., 2006). The combination of our IRD data with other records from Eirik Drift as well as other subpolar North Atlantic sites allowed us to present a comprehensive reconstruction of the transition from the MCA to the LIA. This study reveals the importance of ice discharges in modifying surface circulation in the SPG, as a driver of oscillations in climatic patterns and deep water production in the past, and perhaps again in the near future.

2. Geological and oceanographic setting

Site GS06-144-03 (57.29° N, 48.37° W, 3432 m water depth) is located in the southern tip of Greenland at the Eirik drift (Fig. 1). The site is placed in the northwest part of the SPG, a very sensitive area to climatic and oceanographic changes given that the upper North Atlantic deep water forms in this region (Schmitz and McCartney, 1993). The SPG boundary currents are formed by the North Atlantic current (NAC), the Irminger current (IC), which is the western branch of the NAC and flows towards Greenland, the East Greenland current (EGC) and the Labrador Current (LC) (Fig.1). The IC brings warm and high salinity water to the Labrador Sea whereas the EGC and LC transport colder and lower salinity water and usually carry icebergs and sea ice from the Arctic area.





Oscillations in the amount of ice transported by the EGC and LC may result in freshening of the
SPG affecting the strength of SPG circulation. Fluctuations in the SPG circulation have been
suggested as the driver of oscillations in decadal deep water production and climate variability in
the North Atlantic and surrounding continents (Böning et al., 2006; Hakkinen and Rhines, 2004;
Hatun et al., 2005). Two states of equilibrium have been described depending on the strength of the
SPG circulation (Born and Stocker, 2014), when the circulation is strong, more salty water is
advected to the center of the gyre favouring deep water formation in this area, whereas when the
circulation is weak more salty water is advected northeastward to the Nordic Seas and the SPG
water gets fresher, which prevents deep convection in the Labrador Sea. However, some increased
convection may occur in the Irminger Basin and Nordic Seas, counterbalancing the lack of
Labrador Sea convection. Changes in the dynamics of the SPG are mainly driven by cyclonic winds
and buoyancy forcing (Born and Stocker, 2014), therefore, freshwater input via iceberg discharges
may be a critical factor modifying the circulation in the SPG and the deep water formation in the
Labrador Sea.

3. Materials and methods
Sediments from Site GS06-144-03 were drilled using a multicore device during a cruise on the R/V
G.O. Sars (Dokken and Ninnemann, 2006). A robust chronology has been developed based on 12
AMS radiocarbon dates and $^{210}$Pb measurements at the top of the core (Kleiven et al.in prep.).
Samples were taken every 0.5 cm and the high sedimentation rate at this site allows us to
reconstruct the ice-rafting history of the past 1200 yr at a decadal-scale resolution (mean
sedimentation rate of 0.029 cm/yr, on average ~17 yr between samples). Sediment samples were
sieved using a 63 µm mesh with de-ionized water to eliminate clays and subsequently dried in an
oven. Then samples were dry sieved to extract 63-150 µm fraction which was used in this study to
examine IRD content. This size fraction is coarse enough to be delivered to the open ocean
primarily by drifting ice rather than wind or currents (Fillon et al., 1981; Ruddiman, 1977), yet
lends itself to detailed petrographic analysis (Bond and Lotti, 1995).
Each sample was split with a microsplitter to obtain an aliquot with about 200 IRD grains. The
aliquots were placed in a transparent gridded tray and counted using a high magnification
stereomicroscope which incorporates a light source from the bottom, similar to the transmitted light,
and a light source from the top which emulates reflected light. Using aliquots in a transparent tray
instead of smear slides offers the possibility of moving the grains independently, thus allowing for a
better identification. Additionally, the use of a transparent tray is a key factor to improve the





identification of quartz and feldspar hematite-stained grains (HSG) by the introduction of a white
paper below the tray which enhances the contrast between the hematite-stained portion and the rest
of the grain. This technique is similar to that described in Bond et al. (1997), however, the use of
aliquots presents the advantage that IRD concentrations in the bulk sediment can be calculated to
obtain the total number of IRD (and IRD types) per gram of bulk sediment. A minimum of 200
grains were counted in each sample and the calculated errors for the replicated samples are below
3.2 %. The identification of different groups of minerals such as HSG of quartz and feldspar,
unstained quartz and feldspar, and brown and white volcanic glass (VG) allows us to calculate the
relative abundance of each type of IRD which may be useful to identify the sources of the drifting
ice that transported the IRD (e.g. Alonso-Garcia et al., 2013; Bailey et al., 2012).
SEM x-ray diffraction was performed on selected grains with an energy dispersive spectroscopy
(EDS) equipment at the facilities of the College of Marine Science (University of South Florida).
The EDS equipment used is an EDAX x-ray microanalysis system with an Apollo 10 silicon drift
detector.

4. Results
The total concentration of IRD (Fig. 2) ranges from ~9,000 to 116,000 grains per gram of sediment
(grains/g) which means that icebergs and sea ice reached the studied area during the entire interval
examined in this work. The highest peak of IRD concentration was reached at the end of the MCA
(1169 yr AD) and the intervals with higher IRD concentration occurred approximately at 1000-
1100, 1150-1250, 1400-1450, 1650-1700 and 1750-1800 yr AD with mean values above 50,000
grains/g. The first two of these five intervals of high ice-rafting occurred during the MCA, whereas
the other three intervals of high IRD concentration took place during the LIA.
Volcanic glass (VG) is one of the main components of the IRD with relative abundances up to 59 %
(Fig. 2). This group includes brown VG fragments, usually not vesicular, and white VG fragments,
very light and often with vesicular aspect. The concentration of the total VG shows a similar pattern
to the total IRD concentration with higher values during the same intervals (Fig. 2). The relative
abundance of VG shows higher values during the intervals of higher total IRD concentration. The
relative abundance of white VG is generally lower than 20 % and does not show clear periods of
higher abundance that can be correlated to the record of volcanic eruptions (Gao et al., 2008).
HSG relative abundance ranges between 2 and 30 %, reaching higher values than those observed at
MC52 in the Eastern North Atlantic (Fig. 3, Bond et al., 2001). The record of HSG concentration



shows a different pattern from the total IRD and VG records, with higher concentration from 1400
to 1900 yr AD (Fig. 2). The relative abundance of HSG is also higher after 1400 yr AD, with mean
values increasing to over 15 % from near 5% before 1400 yr AD. This range of variability is
comparable to previous observations across the Atlantic in the late Holocene (Bond et al., 1997;
198 2001).

Among the selected grains to perform x-ray analysis we separated a group of black unclassified
minerals. According to the SEM x-ray diffraction analysis, those grains are mainly composed by
carbon, and we interpreted them as coal fragments. Those minerals occurred in higher abundance
during the MCA and the end of the LIA.

5. Discussion
5.1. IRD sources and significance
The mineralogy found at Site GS06-144-03 suggests several lithological sources for the IRD which
may be associated with icebergs or sea-ice originated from different areas. Volcanic rocks outcrop
mainly in Iceland and the Geikie Plateau area on the East Greenland coast, surrounding Denmark
Strait (Bailey et al., 2012; Henriksen et al., 2009). Volcanic glass can also be atmospherically
transported after volcanic eruptions and be ultimately incorporated in the ice as it has been shown in
Greenland ice core records (Grönvold et al., 1995). This is very likely the case of the white VG
fragments found in our record because our counts of white VG (Fig. 2) do not suggest the presence
of any discrete layer that could be associated with any dated Icelandic eruption (Gao et al., 2008).
This type of IRD was probably deposited on the top of glaciers and sea-ice near Iceland and the
East Greenland coast and then transported in the ice through the EGC. Although some of those
volcanic shards ejected to the atmosphere could have fallen directly in the sea, the preferentially
eastward dispersal pattern of Icelandic tephra following the predominantly westerly winds in the
stratosphere (Lacasse, 2001) argues against the hypothesis of volcanic glass transported by winds to
the study site. Moreover, previous studies suggested the significantly low amounts of tephra
transported towards Greenland prevent finding layers that can be associated with volcanic eruptions
(Jennings et al., 2014). After detailed geochemical studies Jennings et al. (2014) could not recognise
any specific layer that could be used as a tephrochronological event in the SE Greenland coast
during the last millennium. Brown VG fragments are generally solid and not vesicular, suggesting
that they are not windblown shards and were more likely to have been incorporated in the ice from
outcrops in Greenland and Iceland. Similar brown VG fragments were described in
Kangerdlugssuaq trough sediments and were interpreted as coming from the glaciers and sea ice





from the Geikie Plateau area, based on mineralogical and x-ray diffraction analysis data (Alonso-
Garcia et al., 2013).
The presence of HSG in Eirik Drift sediments indicates drift-ice coming from NE Greenland and
the Arctic, where red sandstones outcrop (Bond et al., 1997; Henriksen et al., 2009). Most of the
glaciers in NE Greenland and the Arctic develop floating ice tongues in the fjords where semi-
permanent fast-ice hinders the icebergs from drifting. As a result, most of the IRD carried at the
base of the icebergs is deposited in the fjords (Reeh et al., 2001). Our HSG record from the Eirik
Drift shows a significant amount (up to 30%) of this type of IRD. Therefore, despite substantial
deposition of debris within the fjords, the remainder of the drifted ice still carries considerable
amounts of IRD. We suggest that some of that IRD may have been wind-blown to the top of the
glaciers and/or sea ice at the NE Greenland and Arctic coasts and fjords, rather than directly
incorporated in the bottom layers of the glacier. Those grains were then ice-rafted southwards by
the EGC when the ice was released from the fjords. A similar origin was proposed for HSG
deposited at the SE Greenland coast based on a multi-proxy study (Alonso-Garcia et al., 2013). In
this study, periods of higher HSG abundance were associated with strong ice export from the Arctic
via the EGC.
Variations in Arctic ice export show a significantly positive correlation with the wintertime North
Atlantic/Arctic Oscillation (NAO/AO) during the last decades (e.g. Dickson et al., 2000), although
it also depends on the meridional wind components and the position of the atmospheric pressure
centers (Hilmer and Jung, 2000). Indeed, during the "Great Salinity Anomaly" the freshwater and
ice input to the subpolar North Atlantic happened during a NAO negative phase (Dickson et al.,
2000). Additionally, Darby et al. (2012) demostrated that the sources of Arctic sea ice may change
following the AO and, therefore, we can observe changes in the mineralogy transported by the ice
in sediment cores influenced by the EGC. During the negative state of the AO a strong high
pressure system dominates the Beaufort Sea restricting the Trans-Polar Drift to the Siberian side of
the Arctic Ocean (Mysak, 2001; Rigor et al., 2002) which would bring drift-ice with HSG from the
areas of Severnaya Zemlya and Franz Josef Land. The increase in HSG relative abundance and
concentration at Eirik Drift after 1400 yr AD (Fig. 3) coincides with the shift from positive to
negative NAO conditions reconstructed using tree rings, speleothems and lake records (Olsen et al.,
2012; Trouet et al., 2009). This switch in NAO conditions may have intensified Arctic sea ice
export at the beginning of the LIA, leading to the observed increase in HSG. The sedimentary
record of Feni Drift (Bond et al., 2001), in the NE Atlantic, also shows an increase in HSG relative
abundance during the LIA (Fig. 3). Furthermore, another proxy, the sodium (Na+) concentration in



the Greenland ice core GISP2 (Meeker and Mayewski, 2002) indicates an increase in storminess at
~1400 yr AD. The amount of Na+ in Greenland ice cores has been interpreted as controlled by the
Icelandic Low, and hence, the increase in Na+ may be linked to the NAO negative phase. Enhanced
storminess favours the transport of icebergs and sea ice through the EGC as well as the deposition
of HSG in the sea ice and on top of glaciers, and both processes increase the amount of HSG
transported to Eirik Drift. Greenland temperature also shows a decreasing trend after ~1400 yr AD,
coinciding with the shift to predominantly negative NAO (Kobashi et al., 2010). Colder
atmospheric temperatures and the increase in ice drifted from the Arctic may have contributed to
decrease subpolar sea surface temperature in the subpolar area, favouring icebergs to reach areas
further south such as Feni Drift (Bond et al., 2001).
Coal bearing sediments are present at many areas around the Arctic such as Siberia, Northern
Canada, Greenland and Scandinavia (Polar Region Atlas, 1974; Petersen et al., 2013) and contribute
to high-latitude IRD deposition (Bischof and Darby, 1997; McManus et al., 1996). Even though the
percentage of coal fragments is rather low at our study site (under 5 %, see Fig. 2) the higher
abundance of coal fragments in the Labrador Sea during the MCA may be related to an increase in
drift-ice from the Canadian Arctic during the positive state of NAO/AO. However, these fragments
might also indicate human-related activity which increased in the area during the MCA. Further
analysis should be performed to assess the linkage of those grains to any specific source.
Regardless of the mineralogy of the grains, it is noteworthy the high number of lithics per gram of
sediment recorded in several samples during the MCA (Fig. 2). A recent comprehensive study of
the last 2 millennia (PAGES 2k Consortium, 2013) shows this interval presented sustained warm
temperatures from 830 to 1100 yr AD in the Northern Hemisphere, including the Arctic region. The
high occurrence of IRD from 1000 to 1250 yr AD suggests that during the MCA either a substantial
amount of icebergs drifted to the study area or the drifting icebergs contained considerable amounts
of IRD, or a combination of both explanations. Several studies on East Greenland glaciers and
fjords point to the consistent relationship between calving rate acceleration and the presence of
warm Atlantic water in East Greenland fjords, brought by the Irminger current (Andresen et al.,
2012; Jennings and Weiner, 1996). Warm atmospheric temperatures as well as the presence of
Atlantic water prevent the formation of sea ice in the fjords and in front of the glacier, thus
increasing the calving rate by destabilizing the glacier tongue (Andresen et al., 2012; Murray et al.,
2010). When tidewater glaciers are released from the sea ice, their speed increases due to the
decreased flow-resistance and increased along-flow stresses during the retreat of the ice front, and
rapid changes may be observed in calving rates in response to disequilibrium at the front (Joughin et





al., 2008). At present, Kangerdlugssuaq and Helheim glaciers, located in the central East Greenland
coast, represent the 35 % of East Greenland´s total discharge (Rignot and Kanagaratnam, 2006). If
conditions during the MCA were similar or warmer than at present, the calving rates of these
glaciers may have been even higher than at present, delivering vast amounts of icebergs to the EGC,
where they would release IRD as they melted. Moreover, during the MCA it is likely that other
fjords, such us Nansen and Scoresby Sund, were also ice free during the summer, allowing them to
contribute considerable numbers of icebergs to the EGC. The massive diamicton found in Nansen
fjord sediments between 730 and 1100 yr AD demonstrates that there was continuous iceberg
rafting due to warmer conditions (Jennings and Weiner, 1996). In this context, we postulate that
warm temperatures were the driver of the increased iceberg calving at Greenland fjords and the high
accumulation of IRD at Eirik Drift during late MCA.
After 1250 yr AD several spikes of high IRD abundance occurred during the intervals 1400-1450 yr
AD, 1650-1700 and 1750-1800 yr AD (Fig. 2). Because those intervals occurred within the LIA and
under cold conditions the trigger of iceberg production must have been slightly different from the
drivers proposed for the MCA ice-rafting events. These intervals of higher IRD accumulation
during the LIA are characterized by slightly lower relative abundance of HSG and higher relative
abundance of volcanic grains and other fragments. This points to an intensification of SE Greenland
production of icebergs during the LIA intervals of enhanced ice-rafting. Therefore, for the LIA
events, we advocate for the same mechanism that was put forward to explain rapid releases of
icebergs in Denmark Strait during the last 150 yr (Alonso-Garcia et al., 2013). During cold periods
sea ice becomes perennial along the Greenland coast blocking the seaward advance of glaciers and
hindering icebergs from calving, thus leading to the accumulation of ice mass in the fjords. Based
on model simulations, when the sea ice opens or breaks, the ice flow at the grounding line
accelerates very quickly, triggering a rapid release of the grounded ice stream (Mugford and
Dowdeswell, 2010). In summary, we propose that the high IRD occurrence during the intervals
1350-1450 yr AD, 1650-1700 and 1750-1800 yr AD very likely corresponds to episodes of rapid
iceberg release from SE Greenland fjords. Interestingly, the timing of these intervals of high IRD
deposition coincides with the events of volcanic-solar downturns described by the PAGES 2k
Consortium (2013).

5.2. Influence of ice-rafting on SPG conditions and climate during the last millennium
Our IRD records have been compared with other paleoceanographic and paleoclimatic records from
Eirik Drift and other subpolar North Atlantic sites to obtain a better picture of subpolar conditions



during the last millennium. The planktic foraminifer $\delta^{18}$O record of *Globigerina bulloides* and the
*Neogloboquadrina pachyderma* sin relative abundance from Eirik Drift (Moffa-Sanchez et al.,
2014a; Moffa-Sanchez et al., 2014b) suggest a cooling episode during late MCA (~1100 yr AD) and
a clear drop in temperature after 1200 yr AD (Fig. 4). The coincidence of these temperature drops
with the increasing trend in total IRD concentration at site GS06-144-03, indicates that the growing
iceberg production at East Greenland fjords, due to the MCA warm conditions, started to cool and
freshen Labrador Sea several centuries before the LIA started. The quartz/plagioclase ratio, a bulk
measure of IRD (Moros et al., 2004), also shows an increasing trend at the end of the MCA at sites
in Denmark Strait (Andrews et al., 2009a; see Fig. 4) and off northern Iceland (Moros et al., 2006)
providing further evidence for the intensification of iceberg calving at this time. Colder winter sea
surface conditions have also been recorded off N Iceland after 1200 yr AD (Jiang et al., 2007)
although the sea ice index indicates the first period of severe sea ice conditions only started at
~1300 yr AD (Massé et al., 2008) when annual SST substantially decreased (Sicre et al., 2008), in
agreement with the Denmark Strait data (Fig.4). The reduction in the relative abundance of the
benthic foraminifer *Cassidulina teretis* between 1000 and 1300 yr AD in Nansen fjord indicates a
weaker influence of Atlantic water at the East Greenland coast (Jennings and Weiner, 1996). This
decline in Atlantic water may be explained by a weakening in the northern branch of the Irminger
current which would have favoured the SST decrease and sea ice formation in Denmark Strait and
North of Iceland (Blindheim and Malmberg, 2005). These authors associated the northern Irminger
current weakening with high pressure over Greenland and northerly winds. Moreover, other proxies
from Denmark Strait indicate the strengthening of N and NW winds after ~1250 yr AD, which led
to progressive presence of sea ice exported from the Arctic during winter and spring (Andrews et
al., 2009a; Andrews et al., 2009b).
The remarkably high Atlantic temperatures recorded during the interval ~950-1100 yr AD (Mann et
al., 2009) may indicate SPG circulation was in the strong mode during that time interval (Fig. 4 &
5). Strong SPG circulation enhances the supply of warm Atlantic Intermediate water to the East
Greenland coast, which promotes calving and, subsequently, increases the ice input in the Labrador
Sea region. Switches from weak to strong SPG circulation may happen naturally due to external or
internal forcings, and these changes are currently a matter of debate because of their influence on
North Atlantic climate (e.g. Hakkinen and Rhines, 2004). According to model simulations,
freshwater input (i.e. ice input) to the SPG may trigger weakening of SPG circulation, and this may
be amplified successively by positive feedbacks resulting in further weakening and freshening of
the gyre due to the attenuation of the Irminger current (Born et al., 2010). Specifically for this time
interval, it is important that the main freshwater source was in SE Greenland and reached directly





the Labrador Sea, because a freshwater input into the Nordic Seas may have driven the opposite
effect (Born and Stocker, 2014). Our IRD record evidences an increase in the amount of ice
transported by the EGC to the Labrador Sea from 1000 to 1250 yr AD. This input of freshwater to
the SPG potentially drove a slowdown of deep convection in this area and weakened the SPG
circulation. A recent study also points to enhanced input of the Labrador current to the Labrador
Sea from ~1000 to 1300 yr AD (Sicre et al., 2014), which indicates calving intensified in SW
Greenland and Baffin Bay regions as well. Probably ice from both sources, East and West
Greenland, directly affected the salinity balance of Labrador Sea water and deep convection in this
region. However, even though the freshwater input started at ~1000 yr AD, the SPG circulation
only started to weaken after ~1250 yr AD, as suggested by a record of deep-sea corals from the NE
Atlantic (Copard et al., 2012). Moreover, our IRD data shows a lag between the first temperature
drops at Eirik Drift and the decrease in ice-rafting (Fig. 4), indicating a possible hysteresis between
SPG weakening and Irminger current slowdown. It seems the SPG entered in the weak mode,
because of the reduced convection, but warm intermediate water remained in the fjords for several
years, allowing continued iceberg calving. Also, the response of calving may be slower, particularly
if SST were relatively warm and the fjords were not perennially covered by sea ice.
As the strength of Irminger current input declined, the areas of Denmark Strait and North of Iceland
cooled, and coastal sea ice became perennial after 1450 yr AD, according to the sea ice index $IP_{25}$
(Massé et al., 2008). The *Turborotalita quinqueloba* $\delta^{18}$O record from Eirik Drift (Moffa-Sanchez et
al., 2014b) indicates a shift to colder summer SST in the SPG after 1400 yr AD (Fig. 4), which
coincides with the increase in Arctic ice export reflected by the HSG, the storminess intensification
(Fig. 3), recorded by the $Na^+$ content in the Greenland ice core GISP (Meeker and Mayewski,
2002), and the shift to negative NAO conditions (Trouet et al., 2009). Planktic $\delta^{18}$O and Mg/Ca
from sites in the Norwegian Sea display an initial decrease in temperature at 1200 yr AD and a
subsequent distinct downward shift at ~1400 yr AD, which suggests not only SST cooling but also a
decline in the stratification of the water column, very likely linked to changes in atmospheric
conditions (Nyland et al., 2006; Sejrup et al., 2010). It is clear that sea surface conditions in the
subpolar gyre were rather different before and after ~1200 yr AD. The freshening of the SPG and
the increase in sea ice along the Greenland and Iceland coasts may have been associated with a
change in atmospheric conditions, deepening the Icelandic Low and intensifying winter circulation
over the North Atlantic, i.e. promoting NAO negative conditions and storminess in the subpolar
area. Model simulations point to the development of frequent and persistent atmospheric blocking
events, induced by low solar irradiance, as one of the main drivers to develop the consecutive cold
winters documented in Europe during the LIA (Moffa-Sanchez et al., 2014a). Atmospheric blocking



events derive from instabilities of the jet stream which divert or block the pathway of the westerly
winds (Häkkinen et al., 2011). These events typically predominate during winter and occur linked
to negative NAO index. The cold SST recorded at the subpolar area during low solar irradiance
periods (Moffa-Sanchez et al., 2014a; Moffa-Sanchez et al., 2014b; Sejrup et al., 2010), suggest that
atmospheric blocking events affected the entire North Atlantic regional climate.

5.3. Implications for LIA origin and Norse colonies
It is worth noting that our IRD record shows two types of ice-rafting events: ice-rafting related to
warm temperatures (during the MCA), and ice-rafting linked to rapid releases of the ice
accumulated in the fjords due to cold conditions (during the LIA). During the LIA, the events of
maximum ice-rafting are closely coupled with the minimum values of solar irradiance (Steinhilber
et al., 2009), particularly with the Wolf, Spörer and Maunder minima (Fig. 5). The reconstruction of
radiative forcing based on solar irradiance and volcanic eruptions (Crowley, 2000) also shows low
values during the main events of high IRD occurrence (Fig. 5). Ice-rafting events tend to happen
during intervals of low solar irradiance and cold temperatures in the SPG, often with also
significantly cold summer SST (Fig. 4). Solar irradiance has been put forward as the main trigger
for the Holocene cold events because low solar irradiance induces an atmospheric reorganization
which produces a situation similar to the NAO negative phase (e.g. Bond et al., 2001). Several
records from the high latitude North Atlantic support this hypothesis, displaying cold temperatures
at times of solar irradiance minima during the last millennium (Moffa-Sanchez et al., 2014a; Sejrup
et al., 2010). Precisely dated records of ice-cap growth from Arctic Canada and Iceland show that
LIA summer cooling and ice growth began abruptly between 1275 and 1300 yr AD, followed by a
substantial intensification at 1430-1455 yr AD (Miller et al., 2012). Those authors pointed to the
high volcanic activity during this interval as the main driver for the atmospheric reorganization.
However, a comprehensive review on the topic proposed that a combination of internal and external
forcings contributed to drive Holocene cold events, including the LIA (Wanner et al., 2011), and
recent modelling studies indicated that the weakening of the SPG circulation was not related to
either solar or volcanic forcing (Moreno-Chamarro et al., 2016).
According to our observations, the increase in Greenland calving during the MCA (Fig. 5) took
place before the ice caps started to grow, during an interval of high solar irradiance, high
temperatures in the Northern Hemisphere, and low volcanic forcing. This indicates that the ice-
rafting events of the MCA were not related to the fluctuations driven by solar-volcanic forcing.
Alternatively, we interpret these events as resulting from the acceleration of calving rates in SE





Greenland glaciers, driven by warm temperatures. We postulate that the increase in calving rates
during the MCA induced a decrease in the Labrador Sea salinity, which may have triggered the
weakening of SPG circulation and reduced convection. A decline in Labrador Sea convection
reduces deep water formation in one of the key areas of the North Atlantic, which weakens the
AMOC, and in turn decreases oceanic heat transport to this area (Born et al., 2010; Moreno-
Chamarro et al., 2016). Once the SPG entered in the weak mode this area received less heat and
became more sensitive to external forcings which may have generated further cooling. This
interpretation is in agreement with recent model simulations which suggest that a weakening of the
SPG circulation could have induced the LIA cooling, and this shift from strong to weak circulation
may have been triggered by freshwater input to the Labrador Sea (Moreno-Chamarro et al., 2016).
Subsequently, low solar irradiance intervals, possibly combined with volcanic emissions, promoted
atmospheric reorganizations which gave rise to prevailing negative NAO conditions and/or
atmospheric blocking events, enhancing cold temperatures in the subpolar area and promoting ice
sheet growth in the Arctic region during the LIA. The development of atmospheric blocking events
in the North Atlantic, as suggested by Moffa-Sanchez et al. (2014a), probably propagated the
atmospheric cooling across Europe and the Nordic Seas. Indeed, the first strong minimum of solar
irradiance associated with the LIA (Wolf, ~1300 yr AD) occurred when the Labrador Sea was
already fresher and SPG circulation was weak (Fig. 5), according to our interpretations and to
Copard et al. (2012) deep-sea corals record. The reconstruction of combined solar and volcanic
forcing (Fig. 5) shows a trend of lower values after 1450 yr AD with a first step of low values
during the Wolf minimum indicating that volcanic forcing may also have played an important role
in modifying the atmospheric conditions. However, we consider that the decrease in Labrador Sea
salinity prior to the Wolf minimum was crucial to produce changes in SPG circulation. Once the
SPG entered the weak mode, the effects of solar and volcanic forcing produced a deeper impact on
North Atlantic climate. It is likely that the LIA would not have been such a cold and widespread
event if the SPG circulation was strong and deep convection was active at the time.
The results of this study can be linked to the expansion and demise of the Norse colonies.
According to historical data the Norse expansion and colonization of Iceland and Greenland
occurred during the warmer climate conditions of the MCA which favoured fishing and farming in
these regions (Kuijpers et al., 2014; Ogilvie et al., 2000; Ogilvie and Jónsson, 2001; see Fig. 3). Our
study indicates that even though calving intensified after the settlement of the Norse colonies in
Greenland, climatic conditions during the late MCA were still favourable because the strong
circulation in the SPG supplied relatively warm water to SE Greenland coast. Therefore, the fjords
were not perennially covered by sea ice and it is likely that a rather continuous calving may have



helped hunting. However, after several decades of intense calving and melting of Greenland
glaciers, the Labrador Sea got fresher and the SPG circulation started to weaken triggering a change
in oceanic and atmospheric conditions. The reduction of deep convection decreased the transport of
heat to the NW subpolar area and enhanced sea ice occurrence in the fjords, which deteriorated the
living conditions in Greenland. The subsequent cooling and increase in storminess brought by the
shift in atmospheric conditions (predominant NAO negative state and increase in atmospheric
blocking events) very likely prompted the abandonment of the Greenland Norse settlements at the
beginning of the LIA (Ogilvie et al., 2000, Fig. 3).

6. Conclusions
Sediments from Eirik Drift were studied in order to examine the variations in ice-rafting during the
last millennium and its linkage to LIA development. IRD in the 63-150 μm fraction show higher
concentration during the intervals: ~1000-1100, ~1150-1250, ~1400-1450, ~1650-1700 and ~1750-
1800 yr AD. The identification of different minerals allowed us to link the IRD with potential
sources and better interpret the ice-rafting events. The main IRD source was along the SE
Greenland coast, although during the LIA the greater concentration and relative abundance HSG
supports an increase in the contribution of ice exported from the Arctic region and NE Greenland
via the EGC. Two different types of ice-rafting events have been recognised: (1) ice-rafting
recorded during the MCA, which we interpret as being related to the acceleration of calving rates in
SE Greenland glaciers driven by warm oceanic and atmospheric temperature; and (2) ice rafting
events during the LIA, which have been linked to rapid releases of the ice accumulated in the fjords
due to the perennial sea ice developed in Greenland coast during cold periods.
The comparison of our IRD records with other North Atlantic reconstructions of ice-rafting, sea
surface and deep ocean conditions provides a better picture of the development of the LIA. We
postulate that the enhanced ice discharge during the MCA, decreased sea surface salinity in the
Labrador Sea, which in turn reduced Labrador Sea convection and weakened SPG circulation. The
reduction in convection in the Labrador Sea, one of the key areas of deep water formation in the
North Atlantic, potentially weakened the AMOC and decreased oceanic heat transport to the high
latitudes, particularly to the Labrador Sea region. Reduced convection also diminished the arrival of
warm water from the NAC to SE Greenland coasts inducing perennial sea ice occurrence and
cooling the atmosphere which and promoted ice sheet growth in the Arctic. Cooling and freshening
of the SPG preconditioned the subpolar area to be more sensitive to external forcings. Therefore,
the subsequent atmospheric and oceanographic reorganizations induced by solar and volcanic





forcing generated extremely cold conditions in the North Atlantic during the LIA, through a shift to
predominantly NAO negative conditions and the development of atmospheric blocking events in the
North Atlantic. These events boosted further cooling across Europe and the Nordic Seas. The
combination of a fresher SPG with the solar-volcanic induced atmospheric change generated harsh
conditions in the North Atlantic which caused the abandonment of the Norse colonies in Greenland
around 1400 yr AD.
This study puts forward the idea that the development of the exceptionally cold conditions of the
LIA may be better explained by the previous freshening of the Labrador Sea due to enhanced ice-
rafting during the MCA and the subsequent weakening of the SPG circulation. This finding may be
fundamental to model future climate conditions given that calving in the SE Greenland glaciers has
been increasing during the last decade.


Acknowledgements. This project was funded by NSF grants OCE-0961670 and OCE-1258984, and
the Comer Science and Education Foundation grant CP75. Tony Greco is acknowledged for
analytical support with the SEM analysis. MAG would like to acknowledge the support from A.E.
Shevenell, J. Dixon and D. Hollander during her postdoc at USF, and funding from Portuguese
National Science and Technology Foundation (FCT) through the postdoctoral fellowship
SFRH/BPD/96960/2013 and CCMAR funding UID/Multi/04326/2013.

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





Figure Captions
Figure 1. A) Location of multicore GS06-144-03 (red star) and other sites in the Northern North
Atlantic whose records have been used to support the hypothesis proposed in this work. General
North Atlantic circulation is shown according to Schmitz and McCartney (1993). The location of
Norse settlements in Greenland is shaded and indicated with ES (Eastern settlement) and WS
(Western settlement). B) Temperature and salinity profiles of the first 1000 m at site GS06-144-03
obtained though Ocean Data View (http://odv.awi.de/en/home/) from the World Ocean Atlas 2013
(Locarnini et al., 2013; Zweng et al., 2013).

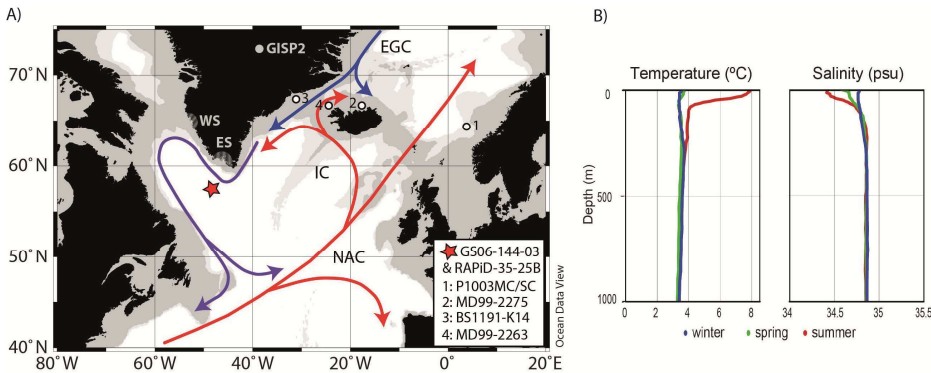







Figure 2. Ice-rafted debris (IRD) records from site GS06-144-03. a) Coal grains relative abundance;
b) Hematite stained grains (HSG) relative abundance; c) total volcanic glass (VG) relative
abundance (brown line) and  white VG relative abundance (shaded area); d) total IRD concentration
in each sediment sample (black line), and IRD concentration not including the white volcanic glass
(shaded area); e) concentration of HSG; f) concentration of total VG (brown line) and white VG
(shaded area); g) Northern Hemisphere sulphate aerosol injection by volcanic eruptions (after Gao
et al. (2008), revised in 2012). Blue horizontal lines indicate mean values for the intervals they
encompass. The approximate standard duration of the Little Ice Age (LIA) and Medieval Warm
Period (MWP) has been shaded in blue and red respectively.

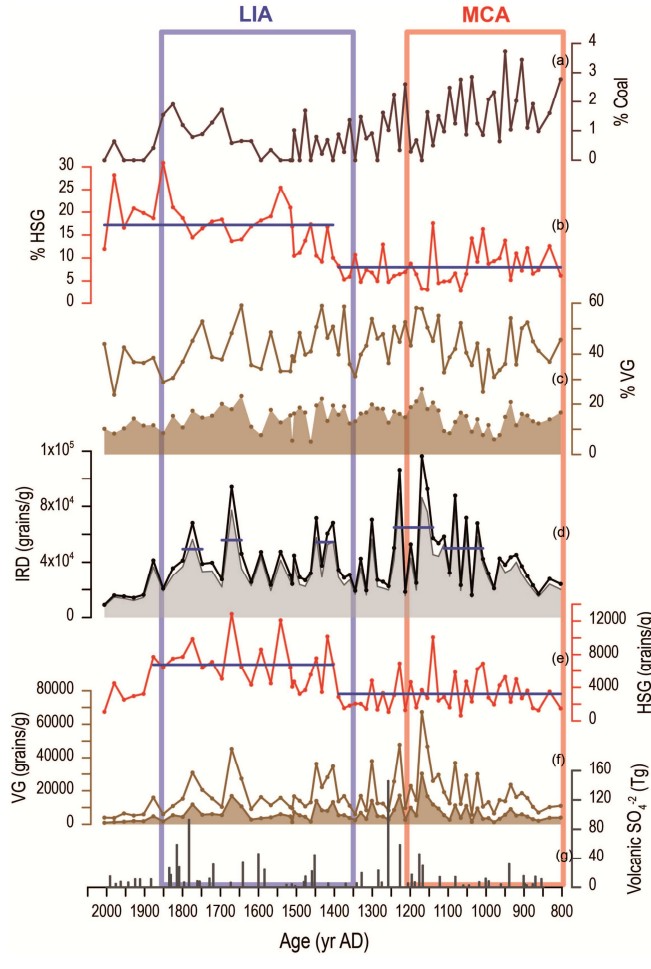




Figure 3. LIA shift at ~1400 yr AD (green vertical bar) in several records compared to site GS06-
144-03 IRD records. a) Winter NAO index reconstructions by Trouet et al. (2009, grey line) and
Olsen et al. (2012, black line); b) Greenland surface temperature reconstruction of the last
millennium (Kobashi et al., 2010); c) Na+ record from GISP2 (Meeker and Mayewski, 2002); d)
HSG record from Eirik Drift (red line)and from Feni Drift in the NE Atlantic (black dashed line,
Bond et al., 2001); e) total IRD concentration; f) HSG concentration. The main events in Norse
colonisation and abandonment of settlements are depicted on the top of the figure , according to
Ogilvie et al. (2000).

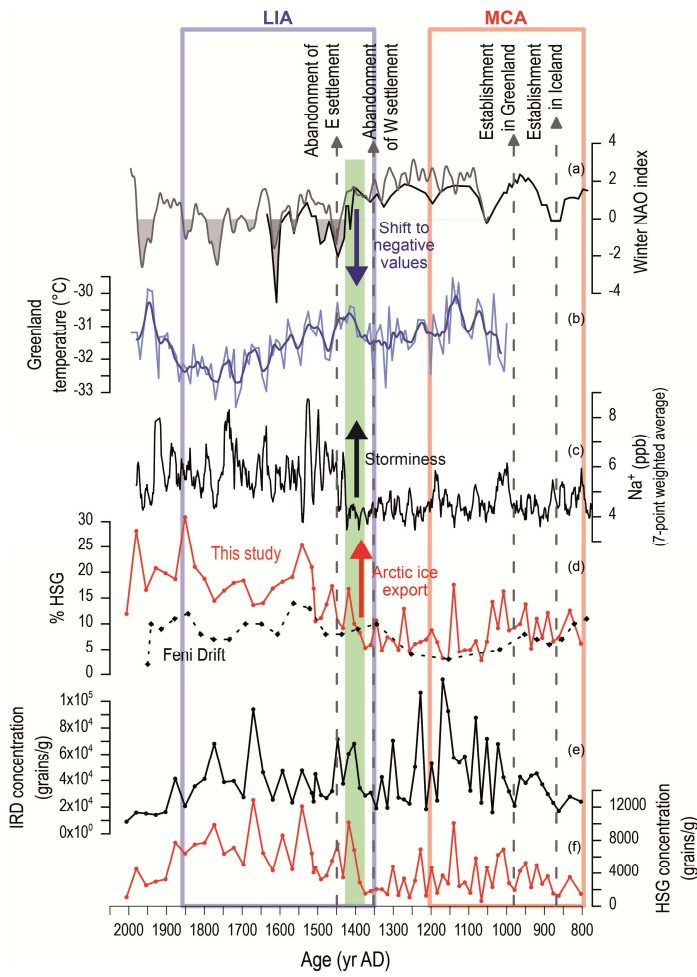




Figure 4. Comparison of IRD records from site GS06-144-03 with subpolar North Atlantic records
of sea surface temperature, ice-rafting and sea ice. a) Atlantic Multidecadal Oscillation (AMO) SST
anomaly (Mann et al., 2009); b) N. pachyderma dex δ18O record from the Norwegian Sea (Sejrup
et al., 2010), c) T. quinqueloba δ18O record from site RAPiD-35-25B at Eirik Drift; d) HSG
relative abundance from site GS06-144-03 (solid line, this study) and from Feni Drift (dashed line,
Bond et al., 2001), e) Sea ice index (IP25) from site MD99-2275, NW of Iceland (Massé et al.,
2008), f) Diatom-based winter SST from site MD99-2275 (Jiang et al., 2007), g) Relative
abundance of the Atlantic waters indicator Cassidulina teretis from Nansen Fjord (Jennings and
Weiner, 1996), h) Relative abundance of N. pachyderma sin from Eirik Drift (Moffa-Sanchez et al.,
2014b), i) G. bulloides δ18O from Eirik Drift (Moffa-Sanchez et al., 2014a), j) Quartz vs
plagioclase ratio, a proxy for ice-rafting, from MD99-2263 (Andrews et al., 2009), k) total IRD
concentration from site GS06-144-03 (this study). Grey vertical bars indicate the periods in which
IRD concentration is higher at site GS06-144-03.





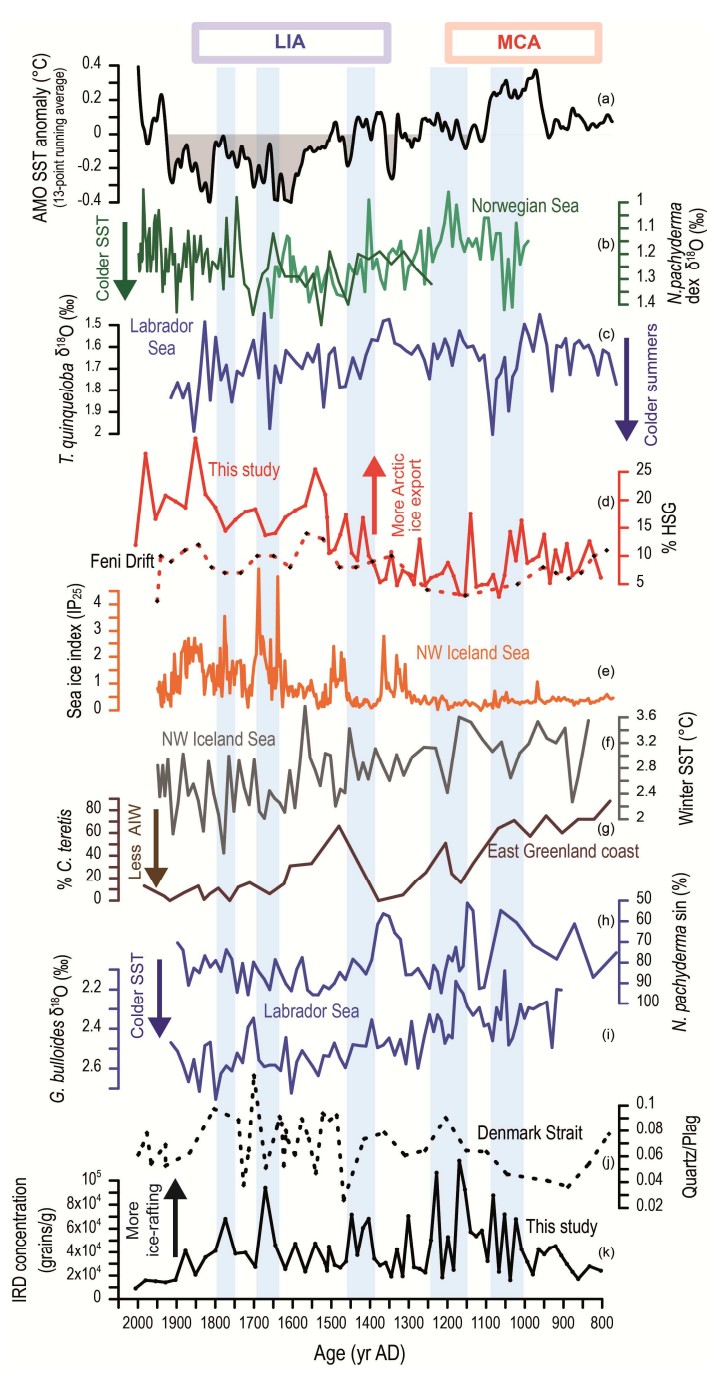




Figure 5. Sequence of events during the transition from the MCA to LIA and correlation to the
potential forcings. a) Winter NAO index reconstructions by Trouet et al. (2009, grey line) and Olsen
et al. (2012, black line); b) Atlantic Multidecadal Oscillation (AMO) SST anomaly (Mann et al.,
2009); c) total volcanic glass (VG) relative abundance at site GS06-144-03; d) total IRD
concentration at site GS06-144-03; e) Reconstruction of total solar irradiance based on 10Be
isotopes from ice cores (Steinhilber et al., 2009); f) Net radiative forcing based on solar irradiance
and volcanic eruption reconstructions (Crowley, 2000). During the interval shaded in red SPG
circulation was stronger, according to the interpretations of this work, whereas during the interval
shaded in blue SPG circulation was weak. The letters in the solar irradiance record indicate the
minima of solar irradiance named Oort (O), Wolf (W), Spörer (S), Maunder (M) and Dalton (D).

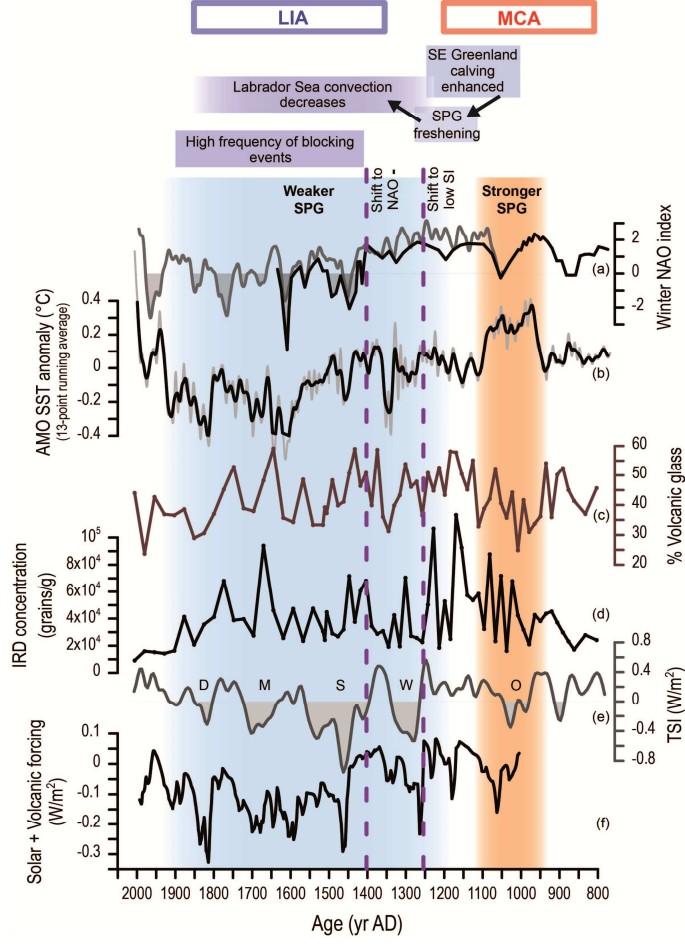
