# Peer review of "M. Alonso-Garcia1,2,3\*, H. F. Kleiven4, J.F. McManus5, P. Moffa-Sanchez6, W. Broecker5 and"

_Climate of the Past, 2016_

## Referee Comment (RC1) · Anonymous Referee #1 · 19 Aug 2016

The paper by Alonso-Garcia and co-authors presents a high resolution record of ice-rafting in the Labrador Sea during the past millennium that allows assessment of the effect of freshwater discharges on the North Atlantic circulation for the first time. Several periods with relatively high debris concentration are identified in this record, periods that extend both the Medieval Climate Anomaly (MCA) and the Little Ice Age (LIA), with debris origin suggested from SE Greenland and the Arctic region respectively. The authors, in addition, compare this new record with other climate reconstructions from the subpolar North Atlantic, and hence argue, first, that a warm medieval climate might have enhanced iceberg calving along the SE Greenland coast, freshening the subpolar gyre region, and later, that this freshening could have forced a weakening in the subpolar gyre/North Atlantic circulation through reduced Labrador Sea oceanic deep convection, itself leading to reduced northward oceanic heat transport and, eventually,

to the cold conditions in the North Atlantic during the LIA.

In my opinion, the result of this paper could be of great interest for the community and, thus, worth publication. I found the paper mostly clear and well written. I have, nonetheless, some concerns about the interpretation of the records (see below) that I would like the authors to address before I can recommend publication. Since it might require important changes in the paper, I suggest major revisions.

The role of the North Atlantic Oscillation (NAO): Throughout the entire manuscript, the authors argue that the NAO could potentially have played a key role in driving the Arctic ice export to the Labrador Sea. This interpretation is based on the Trouet et al. [2009]'s NAO reconstruction, which exhibits a marked shift from persistent positive phases during the MCA to more variable phases in the LIA that agree with the reconstructed increase in the percentage of hematite-stained grains in this study. The robustness of this NAO reconstruction, however, was put into question in Lehner et al. [2012]; and, more importantly, it was updated in Ortega et al. [2015], using a larger amount of proxies and a more robust reconstruction technique. This new reconstruction shows more positive NAO phases for the period ca. 1150–1400 CE, probably associated with the strong volcanic activity during these years; it does not show, however, the strong NAO shift any longer.

Additionally, the authors find results of this record in agreement with the modelling study by Moreno-Chamarro et al. [2016]; in this study, in fact, most of the reconstructed changes in upper-ocean temperature and salinity, in sea ice conditions, or in wind field during the LIA, are explained by an abrupt weakening in the SPG alone, without invoking the NAO at all.

I therefore wonder the need to explain results from this study in terms of the NAO, if the connection might actually not be so clear, and if previous study have already found that changes could be driven by the SPG alone – of course, it was a modelling study, and it is said that all models are "wrong", but the study here under review is indeed

supporting it so, why not building upon it?. For these reasons, I strongly suggest the authors to rethink the interpretation of their results.

Minor comments:

Abstract

L20 – Add comma before "modifying"

L24 – Is the core's exact geographical information, depth and complete name really needed in an Abstract? I would suggest a better use for the few Abstract words we usually have

L25 – "IRD . . . shows"

L25 – "higher" than? This happens several time throughout the manuscript. I suggest "relatively high", for example, since a reference level is not defined to separate high concentrations from not so high

L30 – Acronym HSG is not used in the Abstract

L32 – What do you mean by "cooling events during the LIA"?

L36 – "internal feedbacks" is here a bit vague. I do not exactly see what feedbacks we should think of

L37 – "Atlantic meridional overturning circulation": I would say North Atlantic circulation, without further distinction

L38 – Add comma before "inducing"

L39 – Please, rephrase "volcanic input"

L38–40 – The fact that a cold North Atlantic climate during the LIA might have preconditioned it to be more sensitive to external forcing is not a direct result from this study, but a theory – which I do not see and agree with. It should be removed from the Abstract anyway

1. Introduction

L59, and later on in the text several times too – One should be careful with such a statement. In general, the demise of the Norse Greenland settlements is seen as the result of a bad adaptation to environmental, socioeconomic, and cultural changes occurring at that time. I suggest reading Dugmore et al. [2012], for instance. This sentence, therefore, needs to be softened

L71 – "be the drives of" vs. drive

L76–78 – These results are based on a modelling study and could, therefore, be strongly model-dependent. I would suggest something like: "Freshwater input to this region can potentially reduced oceanic deep convection, slowing down the Atlantic circulation and its related heat transport [e.g., Born et al., 2010; Moreno-Chamarro et al., 2015]". A few notes:

- deep convection and deep water formation in the end refer to the same thing

- oceanic deep convection, in contrast to atmospheric deep convection. It is worth mentioning it at least once at the beginning of the text

- Atlantic circulation better than Atlantic meridional overturning circulation. The former comprises the AMOC, the SPG, etc., which all contributing to the oceanic heat transport

- In Moreno-Chamarro et al. [2016] is was also shown this – but only affecting the SPG

L92–93 – Moreno-Chamarro et al. [2015] clearly showed the exact mechanism that drives such an impact. Worth adding the reference. The word "may" might thus be replace by "can"

2. Geological and oceanographic setting

L127 – Add comma after Labrador Sea

L128 – Please, rephrase "usually". In fact, the East Greenland Current transports about 90% of the total sea ice that goes out of the Arctic. It is hence more than usually

3. Materials and Methods

L170 – Add comma before which

4. Results

L181–191 – Here "higher" without giving any reference is used several times

L191 – Have you tried the new reconstruction of volcanic aerosols in Sigl et al. [2015]? They have a better constrained of the eruption's timing plus distinguishing between North Hemisphere, Tropical, and South Hemisphere eruptions

5. Discussion

L243 onward – Here, the NAO discussion begins

L244 – Is there a more updated reference than Dickson et al. [2000] that shows this connection?

L244 – I have a problem when you treat the Arctic Oscillation (AO) and NAO identically. Although the AO and NAO correlate, especially in winter, they are not identical. In this paragraph AO and NAO are treated as if they were interchangeable

L246 – Then, if such a strong event can occur under a negative NAO phase, why do we need the previous statement? These two sentences contradict each other, and in fact seem to suggest that a strong Arctic freshwater export (also sea ice) can occur under any NAO phase. Is that what you here mean? Does positive correlation here mean that a positive NAO phase leads to more export? Or less, because it is southward, hence negative? This is very confusing. Please, clarify (see above about the NAO role, anyway)

L262 – "and hence" I do not see the causal connection. Since it is an interpretation

of the proxy, shouldn't it be instead something like "... Icelandic Low in a way that the increase in ..."

L305 – Add comma after "conditions"

L307 – "higher"

L320 – Please, rephrase "volcanic-solar downturns"

L327 – "sin" should also be italics, right?

L329 onward – Please, add the letter of the time series you refer after each "Fig. 4". This Figure contains a lot of information and it is difficult to find what you mean from the name and without any further help

L331 – This is interesting: would the freshwater input from these iceberg be large enough to trigger such a change? Is there a way to get an estimate?

L335–339 – From the text, it is not clear whether this climate proxies support your conclusions. Please rephrase and clarify

L339 – "Denmark Strait data". Which one?

L345 – Rephrase "moreover" ("In agreement with this"?)

L345 – "climate proxies". Proxy per se does not mean much

L348 – Could you please summarize the main finding of all these climate reconstructions in some sentences?

L349 – Rephrase "remarkable" (anomalously?)

L358 – This is also found in Moreno-Chamarro et al. [2016] in the context of the last millennium

L359 – Actually, what matters then is that the freshwater input is in the Labrador Sea or in the Nordic Seas. The source may be irrelevant (SE Greenland or higher latitudes),

as long as it affects deep water formation. Please, rephrase

L364 – "Labrador Current", "Irminger Current", etc., like in Labrador Sea, Irminger Sea, etc.

L371 – I do not understand why this is an hysteris problem. Please, clarify

L373 – This is interesting to point further: usually climate models that simulate past climates do not have enough resolution to characterize this sort of mechanism and, generally, do not put freshwater input from Greenland melting into the ocean. If the mechanism here proposed was actually at play, then the model might be missing a relevant source of freshwater that can potentially drive relevant climate changes, like the LIA. It is worth adding to the Discussion.

L385 – A more stratified water column also results from the upper-ocean freshening, because this reduces the seawater density, stopping convection. Such freshening can result from an increase of the Arctic freshwater export and from a reduced salt transport by the SPG [Moreno-Chamarro et al., 2016]

L387 – "SPG"

L389–391 – Under negative NAO, the Icelandic Low is actually not so deep with a weakened winter circulation over the North Atlantic. Hence, it is more wavy and prone to atmospheric blocking situations

L404 – Is it possible to talk about "closely coupled" with the temporal resolution of the record presented here?

L406 – There are already new reconstructions of volcanic eruptions and solar variability. Crowley [2000] is an out-of-date version, even for the CMIP5

L409 – Maybe for other cold events in the Holocene, solar irradiance did indeed play a big role. For the LIA cooling, newest works suggest a dominant role of the volcanic forcing instead [e.g. Atwood et al. 2016]

L418 – "internal and external forcings" should be external forcings and internal climate variability

L424 – Please, rephrase "low volcanic forcing"

L431 – "which weakens the North Atlantic circulation" including AMOC, SPG, etc.

L438 – Here again the authors argue about the role of the NAO, having cited two lines before the work of Moreno-Chamarro et al. [2016], who actually found no role of the NAO in the LIA onset

L443 – "first strong minimum of solar irradiance associated with the LIA (Wolf, ∼1300 yr AD)" The actual timing of the LIA defers very much in the literature, but it is usually given around AD 1450–1550. AD 1300 is actually rather soon for the LIA

L450–452 – This statement strongly needs a citation. If it is not the result of previous studies but a theory here proposed, then it should be rephrased to make clear that it is so, also suggesting some physical mechanism to support it

L457 – Please, add comma before "even though"

L462 – Please, add comma before "triggering"

L467 – See above about Norse Greenland settlements

6. Conclusions

L472 – "shows" "higher"

L485 – Remove comma after "MCA"

L488 – Change AMOC for North Atlantic circulation – a weak SPG also transports less heat

L489 – The arrival of less warm waters is essentially the reduced heat transport to the Labrador Region (and SE Greenland) from the previous sentence

L491 – Please, rephrase "which and promoted"

L491 – "Cooling and freshening . . ." is again not clear whether it is a theory. In any way, it is not the result of this paper and should not go in the Conclusions

L496 – It is not a result of this paper how atmospheric blocking events boosted further cooling across Europe and Nordic Seas. It should hence not go in the Conclusions

L497 – "solar-volcanic-induced"

L498 – Again, the Norse issue

L504 – This sentence needs a citation

Here, I strongly encourage the authors to combine Discussion and Conclusions into one single section

Figure 5:

L771 – "correlation". Let's keep this word for statistical uses. Better, maybe, connection?

References

Atwood, A. R., Wu, E., Frierson, D. M. W., Battisti, D. S., & Sachs, J. P. (2016). Quantifying Climate Forcings and Feedbacks over the Last Millennium in the CMIP5–PMIP3 Models*. Journal of Climate, 29(3), 1161-1178.

Dugmore, A. J., McGovern, T. H., Vésteinsson, O., Arneborg, J., Streeter, R., & Keller, C. (2012). Cultural adaptation, compounding vulnerabilities and conjunctures in Norse Greenland. Proceedings of the National Academy of Sciences, 109(10), 3658-3663.

Lehner, F., Raible, C. C., & Stocker, T. F. (2012). Testing the robustness of a precipitation proxy-based North Atlantic Oscillation reconstruction. Quaternary Science Reviews, 45, 85-94.

Moreno‐Chamarro, E., Zanchettin, D., Lohmann, K., & Jungclaus, J. H. (2015).

Internally generated decadal cold events in the northern North Atlantic and their possible implications for the demise of the Norse settlements in Greenland. Geophysical Research Letters, 42(3), 908-915.

Ortega, P., Lehner, F., Swingedouw, D., Masson-Delmotte, V., Raible, C. C., Casado, M., & Yiou, P. (2015). A model-tested North Atlantic Oscillation reconstruction for the past millennium. Nature, 523(7558), 71-74.

Sigl, M., Winstrup, M., McConnell, J. R., Welten, K. C., Plunkett, G., Ludlow, F., ... & Fischer, H. (2015). Timing and climate forcing of volcanic eruptions for the past 2,500 years. Nature.

---

## Referee Comment (RC2) · J. Andrews (Referee) · 23 Aug 2016

I read the paper with considerable interest. I would make two comments to start: 1) I have worked on the "upstream" issues of ice-rafting and sediment provenance for nearly 3 decades hence feel reasonable confident to comment on the paper (e.g. Andrews and Jennings, 2014), and 2) I was a co-author on the 2013 Alonso-Garcia et al. paper.

The basic premise behind the paper is that changes in the amount and source of ice-rafted material (IRD) explicitly contain information about changes in the flux of freshwater, hence can potentially provide information on deep water formation. This premise requires that the proxy provides an unambiguous signal linked to freshwater exports, and of course the link is that the IRD is exported to the Erik Drift either in icebergs or

in sea ice.

The paper provides no information on the chronology other than to say it is discussed in a paper that is listed as "in press"—but it is not in the reference list. It is also important, in my view, to state what has been used for the ocean reservoir correction and was an error attached to the value? This issue limits how well the chronology can be defined, hence the reliability of correlations with other records. It is a difficult issue that bedevils all of us (see ref. to Sjerup et al. 2010, their ref list). The authors note that Jennings et al (2014) were not able to identify a specific Icelandic tephra in the last 1 cal ka or so, hence it is difficult to constrain the possible $\Delta$R.

I feel quite strongly that there needed to be more discussion on rationale for choosing the $> 63\mu$m fraction as an IRD signal (Andrews, 2000). I think the only really unambiguous IRD grain-size signal are clasts > 2 mm (Grobe, 1987), although a solid case can be made for a $\geq 250$ $\mu$m. When the fine sand and greater fractions are being identified, especially on a Drift, then I think an initial analysis should include the entire grain-size spectra (Prins et al., 2002) as this, typically, indicates IRD as a distinct hump at the coarse end of the grain-size spectra. I also note that there is no discussion on iceberg history (e.g. (Bigg, 1999; Bigg et al., 2014; Bigg and Wilton, 2014)) or on sea ice, especially the export of the "storis" (Schmith and Hanssen, 2003).

Finally, the discussion of the provenance of the $> 63\mu$m fraction might have usefully identified (on their Fig. 1?) the major tidewater ice streams/glaciers of SE/E/NE Greenland and have referenced the likely annual flux (km3/yr) versus that of sea ice, this would help in trying to establish provenance. For example, coal outcrops in the area of Nansen Fjord, East Greenland, and it has been recorded in sediments on the inner shelf (Jennings, person. Commun. 2010) but I am not sure if this was stated in any of her publications. The issue of the source(s) HSG is an important one given the attention it achieved through Gerard Bond's work. The most probable source is the Devonian outcrop ca 73°N, NE Greenland (Larsen et al., 2008) in the area of Kasjer Franz Joseph Fjord. Several cores were taken from this area during a Polarstern cruise

(Evans et al., 2002; Hubberten and al., 1995; Stein, 2008) although evidence for significant IRD output over the last millennium is muted and the number of tidewater glaciers on the outcrop is limited.

Thus although I have some concerns about the paper I also believe that it represents an important contribution to our understanding of climate change in an area that is critical to our understanding of the Earth's Climate System.

John T. Andrews

References cited:

Andrews, J.T., 2000. Icebergs and Iceberg Rafted Detritus (IRD) in the North Atlantic: Facts and Assumptions. Oceanography 13, 100-108.

Andrews, J.T., Jennings, A.E., 2014. Multidecadal to millennial marine climate oscillations across theDenmark Strait (∼66° N) over the last 2000 cal yr BP. Climate of the Past 10, 325-343.Bigg, G.R., 1999. An estimate of the flux of iceberg calving from Greenland. Arctic, Antarctic, and Alpine Research 31, 174-178.

Bigg, G.R., Wei, H.L., Wilton, D.J., Zhao, Y., Billings, S.A., Hanna, E., Kadirkamanathan, V., 2014. A century of variation in the dependence of Greenland iceberg calving on ice sheet surface mass balance and regional climate change. . Proceedings Royal Society A 470, 20130662.

Bigg, G.R., Wilton, D.J., 2014. Iceberg risk in the Titanic year of 1912: was it exceptional? Weather 69, 100-104.

Evans, J., Dowdeswell, J.A., Grobe, H., Niessen, F., Stein, R., Hubberten, H.-W., Whittington, R.J., 2002. Late Quaternary sedimentation in Kejser Joseph Fjord and the continental margin of East Greenland, in: Dowdeswell, J.A., O'Cofaigh (Eds.), Glacier-influenced sedimentation on High-Latitude conintental margins. Geological Society London, pp. 149-179.

Grobe, H., 1987. A Simple Method for the Determination of Ice-Rafted Debris in Sediment Cores. Polarforschung 57, 123-126.

Hubberten, H.-W., al., e., 1995. Die Expedition ARKTIS-X/2 mit FS "Polarstern" 1994, p. 86.

Larsen, P.-H., Olsen, O., Clack, J.A., 2008. The Devonian basin in East Greenland–A review of basin evolution and vertebrate assemblages, in: Higgins, A.K., Gilotti, J.A., Smith, P.M. (Eds.), The Greenland Caledonides. Evolution of the Northeast margin of Laurentria. Geological Society of America, Boulder, CO, pp. 273-292.

Prins, M.A., Bouwer, L.M., Beets, C.J., Troelstra, S.R., Weltje, G.J., Kruk, R.W., Kruijpers, A., Vroon, P.Z., 2002. Ocean circulation and iceberg discharge in the glacial North Atlantic: Inferences from unmixing of sediment sizes. Geology 30, 555-558.

Schmith, T., Hanssen, C., 2003. Fram Strait ice export during the nineteenth and twentieth centuries reconstructed from a multiyear sea ice index from Southwestern Greenland. Journal of Climate 16, 2782-2791.

Stein, R., 2008. Arctic Ocean Sediments. Processes, proxies, and Paleoenvironment. Elsevier, New York.

---

## Referee Comment (RC3) · Anonymous Referee #3 · 2 Sep 2016

Alonso-Garcia and coauthors present high-resolution sediment data from Eirik Drift off the southern coast of Greenland. They analyze the amount and mineralogy of ice-rafted debris (IRD), covering approximately the last 2000 years. The primary results are several episodes of increased deposition of IRD and an increase in hematite stained grains after about 1400 AD. This leads the authors to suggest that Arctic sea ice export strengthened and the Atlantic subpolar gyre (SPG) weakened. As a consequence, less Atlantic waters were present in Greenland fjords, changing the stability of tidewater glaciers, their calving rates and thus the amount of IRD transported to the core site. The original data is compared with several older records from the same region in the subpolar North Atlantic.

I think this study contains very valuable new data and that it is a good addition to the existing literature (Copard et al., 2012; Moffa-Sanchez et al. 2014a, Moreno-Chamarro et

al., 2016). Text and figures are mostly clear. Previous works are referenced adequately. I do not fully agree with the interpretation of the data, which is rather speculative on several occasions and likely incorrect in some aspects. I will summarize my criticism in 4 major points:

1) Several of the claims made in the discussion are not well supported by the data. In my opinion, the sediment record does not warrant statements about the stability of the calving front of Greenland glaciers. Fjord environments are extremely complex and heterogeneous in their dynamics so that no robust conclusions can be drawn from a single sediment core several hundreds or thousands of miles away. Ice-rafting occurs both during 'cold' and 'warm' episodes, for which two different and only weakly supported explanations are given. Furthermore, whether the MCA and the LIA periods really had a temperature signal in the relevant regions has not been shown in the manuscript. I think they are better characterized as high and low sea ice periods.

2) Throughout the paper, references are made to an outdated reconstruction of the North Atlantic Oscillation (NAO) (Trouet et al., 2009). It has been shown that the method of this reconstruction is flawed and that its results are not trustworthy (Lehner et al., 2012). A new and more advanced method shows very different results (Ortega et al., 2015). It is clear that "some" change in the atmospheric circulation took place at 1400 AD that led to changes in the proxy records (Trouet et al., 2009; Olsen et al., 2012), but that change is probably unrelated with the NAO. I really do not see the necessity to relate the original data of this manuscript with the NAO and thus recommend to remove this link altogether.

3) The delay between the onset of the stronger freshwater forcing at ∼1000 AD and the weakening of the SPG ∼1250 AD, as well as the further lag of 200 years before the cooling of Atlantic waters in the fjords (page 12) are not reasonable and not well supported by evidence. Present-day observations and modeling show that a slow-down of the SPG after weakening the convection in the Labrador Sea takes place within a single season or at most some few years. There is no physical reason to

assume a delay of multiple centuries. I believe the same is true for the cooling of Greenland fjords, although I do not know that for certain. I suggest the authors estimate the energy balance and fluxes to support their claim or remove this part. On a more general note, the authors seem to expect a strict determinism underlying their data, which is probably not correct. In the highly variable North Atlantic and due to the strong positive feedbacks associated with the SPG, abrupt changes can happen suddenly and in response to rather minor forcing pulses (Moreno-Chamarro et al., 2016). Not every wiggle will have an easily identifiable cause.

4) While a weakening of the SPG at about 1400 AD is consistent with findings by Copard et al. (2012), Moffa-Sanchez et al. (2014a) report more frequent changes in the strength of the gyre. How can these views be reconciled? This should be included in the discussion.

minor comments: l 77: These numbers refer to the last interglacial and are not relevant here. I think the mechanism is reasonable and should be kept, but either specify that these percentages are for a different climatic period or remove them.

l 95: Say when this shift occurred.

l 137: Add a reference.

l 146: How can robustness be claimed when the key publication by one of the co-authors has not even been submitted?

l 215: This argument is not convincing. If the westerly winds prevented dust from being transported upwind to the sedimentation site, they would also have avoided dust from reaching Greenland glaciers as well. However, the climatological wind direction is irrelevant here, because volcanic eruptions are usually short-lived events and dust is transported in the direction of wind at that time. It can not be ruled out that a volcano erupted during a week of predominantly easterly winds, even though that may be the less common situation.

l 229: In this paragraph, I was quite confused whether the individual statements referred to glacier ice or to sea ice. Please be more explicit.

l 246: This argument shows that while the small recent variations in Arctic sea ice export correlated with the NAO, large anomalies in export probably have a different cause, like the GSA. Maybe sea ice during the LIA was thicker in the Arctic and so more ice (volume) was exported without a faster flow? This view would be consistent with findings from Miller et al. (2012) and Lehner et al. (2013). Also, I am unsure whether the manuscript discusses the impact of freshwater forcing from sea ice.

l 358: A more detailed study on this effect is Born et al. (2016).

References (not already included in the manuscript): Lehner et al. (2013), Amplified inception of European Little Ice Age by sea ice-ocean-atmosphere feedbacks, Journal of Climate 26, 7586-7602 Born et al. (2016), Transport of salt and freshwater in the Atlantic Subpolar Gyre, Ocean Dynamics (online), DOI: 10.1007/s10236-016-0970-y
* * *

---

## Author Comment (AC1) · 26 Oct 2016

Response to Reviewer#1 on "Freshening of the Labrador Sea as a trigger for Little Ice Age development" by Montserrat Alonso-Garcia et al.

We would like to thank Reviewer#1 for his thorough work reviewing the manuscript and for his insightful comments. We really appreciate the feedback provided by a person with expertise on climate and ocean dynamics since it helped us to improve the manuscript.

In order to provide context to our replies, the referee's comments have been copied below and our replies are preceded by "REPLY". We agree with most of the suggestions to reformulate the text, so below we only included the discussion comments for which we can give an answer.

The paper by Alonso-Garcia and co-authors presents a high resolution record of ice-rafting in the Labrador Sea during the past millennium that allows assessment of the effect of freshwater discharges on the North Atlantic circulation for the first time. Several periods with relatively high debris concentration are identified in this record, periods that extend both the Medieval Climate Anomaly (MCA) and the Little Ice Age (LIA), with debris origin suggested from SE Greenland and the Arctic region respectively. The authors, in addition, compare this new record with other climate reconstructions from the subpolar North Atlantic, and hence argue, first, that a warm medieval climate might have enhanced iceberg calving along the SE Greenland coast, freshening the subpolar gyre region, and later, that this freshening could have forced a weakening in the subpolar gyre/North Atlantic circulation through reduced Labrador Sea oceanic deep convection, itself leading to reduced northward oceanic heat transport and, eventually, to the cold conditions in the North Atlantic during the LIA.

In my opinion, the result of this paper could be of great interest for the community and, thus, worth publication. I found the paper mostly clear and well written. I have, nonetheless, some concerns about the interpretation of the records (see below) that I would like the authors to address before I can recommend publication. Since it might require important changes in the paper, I suggest major revisions.

The role of the North Atlantic Oscillation (NAO): Throughout the entire manuscript, the authors argue that the NAO could potentially have played a key role in driving the Arctic ice export to the Labrador Sea. This interpretation is based on the Trouet et al. [2009]'s NAO reconstruction, which exhibits a marked shift from persistent positive phases during the MCA to more variable phases in the LIA that agree with the reconstructed increase in the percentage of hematite-stained grains in this study. The robustness of this NAO reconstruction, however, was put into question in Lehner et al. [2012]; and, more importantly, it was updated in Ortega et al. [2015], using a larger amount of proxies and a more robust reconstruction technique. This new reconstruction shows more positive NAO phases for the period ca. 1150–1400 CE, probably associated with the

strong volcanic activity during these years; it does not show, however, the strong NAO shift any longer.

Additionally, the authors find results of this record in agreement with the modelling study by Moreno-Chamarro et al. [2016]; in this study, in fact, most of the reconstructed changes in upper-ocean temperature and salinity, in sea ice conditions, or in wind field during the LIA, are explained by an abrupt weakening in the SPG alone, without invoking the NAO at all.

I therefore wonder the need to explain results from this study in terms of the NAO, if the connection might actually not be so clear, and if previous study have already found that changes could be driven by the SPG alone – of course, it was a modelling study, and it is said that all models are "wrong", but the study here under review is indeed supporting it so, why not building upon it?. For these reasons, I strongly suggest the authors to rethink the interpretation of their results.

REPLY

The role of the NAO in past ocean circulation and climate changes is still not very well understood by climatologists, and, therefore, the literature shows many articles with contradictory findings, which may lead to confuse interpretations of paleo-data. Ortega et al. (2015) improved the reconstruction of NAO during the last Millennium using a selection of 48 proxy records validated by model simulations. This article indicates that the previously published NAO reconstruction by Trouet et al. (2009), which shows persistent positive values for the MCA, was biased due to using only 2 proxy records and, thereby, all paleoclimate interpretations supported on this persistent positive NAO may be incorrect too.

In our article, we didn't mean to base all the interpretation on NAO reconstructions. Instead, we just wanted to link our conclusions about oceanic-atmospheric changes to other records of atmospheric patterns like the NAO. Our record shows an increase in the supply of Arctic Sea ice (inferred by the increase in HSG) associated with the enhanced storminess over Greenland (increase in Na+) inferred by Meeker and Mayewski (2002), which indicates changes in the atmospheric conditions in the Arctic and sub-arctic region. Therefore, we thought about a change in either AO or NAO conditions, but, as the reviewer said it is not necessary to invoke the NAO to interpret our results. The regional atmospheric changes inferred with the proxies may or may not be linked to NAO since we don't really know what is happening in the Azores High.

Following the reviewer suggestion we are going to reformulate our interpretation. Instead of referring to NAO, we will refer to changes in the atmospheric conditions in the Polar and Subpolar regions. Besides the findings of Moreno-Chamarro et al. (2016), a recent article about the Great Salinity Anomaly (Ionita et al., 2016) points to a linkage between atmospheric blocking events, Arctic ice export and freshening of the Labrador Sea. These modelling studies support our interpretations about atmospheric changes in the study area and Labrador Sea freshening. The linkage between the subpolar atmospheric conditions and NAO is out of the scope of this paper and, therefore, all the information regarding NAO will be removed. The NAO reconstruction will be also removed from figures 3 and 5.

Minor comments:

Abstract

L32 – What do you mean by "cooling events during the LIA"?

REPLY

We meant the cold episodes that comprised the LIA, because the LIA is not a single event. Within the LIA there are very cold decades and mild decades. This will be rephrased in the new text to clarify it.

4. Results

L191 – Have you tried the new reconstruction of volcanic aerosols in Sigl et al. [2015]? They have a better constrained of the eruption's timing plus distinguishing between

North Hemisphere, Tropical, and South Hemisphere eruptions

REPLY

Even though Sigl et al. (2015) presents a better chronology and more detailed reconstruction the record is very similar to Gao et al. (2008), at least for the major volcanic events. Also during the elaboration of the manuscript, we checked other sources in order to look for more regional eruptions but still there is no clear correlation with any significant eruption and therefore we believe the majority of the grains are transported by ice.

5. Discussion

L244 – Is there a more updated reference than Dickson et al. [2000] that shows this connection?

REPLY

We revised the sentence, and we believe the references (Mysak, 2001; Rigor et al., 2002) may be more suitable for that statement, so we will modify this.

L244 – I have a problem when you treat the Arctic Oscillation (AO) and NAO identically. Although the AO and NAO correlate, especially in winter, they are not identical. In this paragraph AO and NAO are treated as if they were interchangeable

L246 – Then, if such a strong event can occur under a negative NAO phase, why do we need the previous statement? These two sentences contradict each other, and in fact seem to suggest that a strong Arctic freshwater export (also sea ice) can occur under any NAO phase. Is that what you here mean? Does positive correlation here mean that a positive NAO phase leads to more export? Or less, because it is southward, hence negative? This is very confusing. Please, clarify (see above about the NAO role, anyway)

REPLY

[Figure]

Comments to lines 244 and 246: All references to NAO will be removed from the discussion to avoid misunderstandings and to focus on the effects of freshwater export to the subpolar area. The new text will only link our results to changes in the Icelandic Low and/or atmospheric conditions in the Arctic/subarctic regions.

L331 – This is interesting: would the freshwater input from these iceberg be large enough to trigger such a change? Is there a way to get an estimate?

REPLY

I am not a modeller, but I guess it is possible to give estimates of the freshwater transported by icebergs and sea ice based on information from present icebergs and the amount of IRD they transport. However, this calculation may take a lot of time and it may be better suited for a new paper. I would be happy to see a modeller calculating estimates for this, indeed it will be really interesting to see this calculations not only for the LIA but for Heinrich Events or glacial Terminations.

L348 – Could you please summarize the main finding of all these climate reconstructions in some sentences?

REPLY

These references show mineralogical evidence of ice-rafting and compare the ice-rafting frequency with other proxies that indicate the presence of sea ice and low salinity water in the region of Northern Iceland-Denmark Strait. They conclude that ice export from the Arctic is enhanced by the atmospheric conditions very likely related to the Arctic Oscillation.

L371 – I do not understand why this is an hysteris problem. Please, clarify

REPLY

Well, we just observe a lag between SPG weakening and Irminger current slowdown. Maybe the word hysteresis has a different connotation in the reviewer's field of work.

This will be rephrased to avoid misunderstandings.

L373 – This is interesting to point further: usually climate models that simulate past climates do not have enough resolution to characterize this sort of mechanism and, generally, do not put freshwater input from Greenland melting into the ocean. If the mechanism here proposed was actually at play, then the model might be missing a relevant source of freshwater that can potentially drive relevant climate changes, like the LIA. It is worth adding to the Discussion.

REPLY

We agree.

L385 – A more stratified water column also results from the upper-ocean freshening, because this reduces the seawater density, stopping convection. Such freshening can result from an increase of the Arctic freshwater export and from a reduced salt transport by the SPG [Moreno-Chamarro et al., 2016]

REPLY

We agree.

L404 – Is it possible to talk about "closely coupled" with the temporal resolution of the record presented here?

REPLY

For marine paleoproxies I think we can say they have a similar timing and therefore they are closely coupled.

L406 – There are already new reconstructions of volcanic eruptions and solar variability. Crowley [2000] is an out-of-date version, even for the CMIP5

REPLY

Instead of the volcanic+solar activity from Crowley (2000), the Global volcanic forcing

from Sigl et al. (Sigl et al., 2015) has been included in figure 5.

L409 – Maybe for other cold events in the Holocene, solar irradiance did indeed play a big role. For the LIA cooling, newest works suggest a dominant role of the volcanic forcing instead [e.g. Atwood et al. 2016]

REPLY

We will include this in the discussion

L438 – Here again the authors argue about the role of the NAO, having cited two lines before the work of Moreno-Chamarro et al. [2016], who actually found no role of the NAO in the LIA onset

REPLY

The NAO discussion has been removed

L443 – "first strong minimum of solar irradiance associated with the LIA (Wolf, _1300 yr AD)" The actual timing of the LIA defers very much in the literature, but it is usually given around AD 1450–1550. AD 1300 is actually rather soon for the LIA

REPLY

Well, that is why we wrote "associated with" and not "within the LIA". Anyway, we can rephrase this as "prior to the LIA".

L450–452 – This statement strongly needs a citation. If it is not the result of previous studies but a theory here proposed, then it should be rephrased to make clear that it is so, also suggesting some physical mechanism to support it

REPLY

Yes, this is our hypothesis. Based on the available data and our results we suggest a freshening of the Labrador Sea started well before the LIA started and could have been one of the factors triggering the LIA as suggested by the modelling study of Moreno-

Chamarro et al. (2016). However, we are not suggesting this is the only driver of the LIA, but very likely a weak subpolar gyre enhanced the effect caused by other forcings, such as volcanic and solar irradiance.

References

Gao, C., Robock, A., Ammann, C., 2008. Volcanic forcing of climate over the past 1500 years: An improved ice core-based index for climate models. J. Geophys. Res. 113, D23111. Ionita, M., Scholz, P., Lohmann, G., Dima, M., Prange, M., 2016. Linkages between atmospheric blocking, sea ice export through Fram Strait and the Atlantic Meridional Overturning Circulation. Scientific Reports 6, 32881. Meeker, L.D., Mayewski, P.A., 2002. A 1400-year high-resolution record of atmospheric circulation over the North Atlantic and Asia. The Holocene 12, 257-266. Moreno-Chamarro, E., Zanchettin, D., Lohmann, K., Jungclaus, J.H., 2016. An abrupt weakening of the subpolar gyre as trigger of Little Ice Age-type episodes. Clim. Dyn., 1-18. Mysak, L.A., 2001. Patterns of Arctic Circulation. Science 293, 1269-1270. Ortega, P., Lehner, F., Swingedouw, D., Masson-Delmotte, V., Raible, C.C., Casado, M., Yiou, P., 2015. A model-tested North Atlantic Oscillation reconstruction for the past millennium. Nature 523, 71-74. Rigor, I.G., Wallace, J.M., Colony, R.L., 2002. Response of Sea Ice to the Arctic Oscillation. J. Clim. 15, 2648-2663. Sigl, M., Winstrup, M., McConnell, J.R., Welten, K.C., Plunkett, G., Ludlow, F., Buntgen, U., Caffee, M., Chellman, N., Dahl-Jensen, D., Fischer, H., Kipfstuhl, S., Kostick, C., Maselli, O.J., Mekhaldi, F., Mulvaney, R., Muscheler, R., Pasteris, D.R., Pilcher, J.R., Salzer, M., Schupbach, S., Steffensen, J.P., Vinther, B.M., Woodruff, T.E., 2015. Timing and climate forcing of volcanic eruptions for the past 2,500 years. Nature 523, 543-549. Trouet, V., Esper, J., Graham, N.E., Baker, A., Scourse, J.D., Frank, D.C., 2009. Persistent Positive North Atlantic Oscillation Mode Dominated the Medieval Climate Anomaly. Science 324, 78-80.

[Figure]

[Figure]

Figure 2. Ice-rafted debris (IRD) records from site GS06-144-03. a) Coal grains relative abundance; b) Hematite stained grains (HSG) relative abundance; c) total volcanic glass (VG) relative abundance (brown line) and white VG relative abundance (shaded area); d) total IRD concentration in each sediment sample (black line), and IRD concentration not including the white volcanic glass (shaded area); e) concentration of HSG; f) concentration of total VG (brown line) and white VG (shaded area); g) Northern Hemisphere sulphate aerosol injection by volcanic eruptions (after Gao et al. (2008), revised in 2012) and non-sea salt Sulfur from NEEM Greenland ice core (Sigl et al., 2015). Blue horizontal lines indicate mean values for the intervals they encompass. The approximate standard duration of the Little Ice Age (LIA) and Medieval Warm Period (MWP) has been shaded in blue and red respectively.

This figure is just to show the reviewer that even though the resolution is higher in Sigl et al. (2005) the main events occur at the same timing and there is no consistent linkage between higher concentrations of volcanic fragments and volcanic events. The final figure will include only Sigl et al. (2005) volcanic reconstruction.

**Fig. 1.** Revised version of figure 2

[Figure]

Figure 3. LIA shift at ~1400 yr AD (green vertical bar) in several records compared to site GS06-144-03 IRD records. a) SE Greenland April sea ice concentration (Miettinen et al., 2015); b) SE Greenland April se surface temperature (Miettinen et al., 2015);  c) Na+ record from GISP2 (Meeker and Mayewski, 2002); d) HSG record from Eirik Drift (red line)and from Feni Drift in the NE Atlantic (black dashed line, Bond et al., 2001); e) total IRD concentration; f) HSG concentration. The main events in Norse colonisation and abandonment of settlements are depicted on the top of the figure, according to  Ogilvie et al. (2000).

This is the new figure 3, where the NAO references have been removed

**Fig. 2.** Revised version of figure 3

[Figure]

Figure 5. Sequence of events during the transition from the MCA to LIA and correlation to the potential forcings. a) Hematite stained grains (HSG) relative abundance at site GS06-144-03; b) Na+ record from GISP2 (Meeker and Mayewski, 2002); c) Atlantic Multidecadal Oscillation (AMO) SST anomaly (Mann et al., 2009); d) SE Greenland April sea ice concentration (Miettinen et al., 2015); e) total IRD concentration at site GS06-144-03; f) Reconstruction of total solar irradiance based on 10Be isotopes from ice cores (Steinhilber et al., 2009); f) Radiative forcing based on volcanic eruption reconstructions (Sigl et al., 2015). During the interval shaded in red SPG circulation was stronger, according to the interpretations of this work, whereas during the interval shaded in blue SPG circulation was weak. The letters in the solar irradiance record indicate the minima of solar irradiance named Oort (O), Wolf (W), Spörer (S), Maunder (M) and Dalton (D).

This is the new figure 5, where the NAO references have been removed.

**Fig. 3.** Revised version of figure 5

---

## Author Comment (AC2) · 26 Oct 2016

Response to Reviewer#2 on "Freshening of the Labrador Sea as a trigger for Little Ice Age development" by Montserrat Alonso-Garcia et al.

We would like to thank John Andrews for his interest in our work and his insightful comments about ice-rafting issues. In order to provide context to our replies, the referee's comments have been copied below preceded by "RC" and our replies are preceded by "REPLY".

RC

I read the paper with considerable interest. I would make two comments to start: 1) I have worked on the "upstream" issues of ice-rafting and sediment provenance

for nearly 3 decades hence feel reasonable confident to comment on the paper (e.g. Andrews and Jennings, 2014), and 2) I was a co-author on the 2013 Alonso-Garcia et al. paper.

The basic premise behind the paper is that changes in the amount and source of icer-afted material (IRD) explicitly contain information about changes in the flux of freshwater, hence can potentially provide information on deep water formation. This premise requires that the proxy provides an unambiguous signal linked to freshwater exports, and of course the link is that the IRD is exported to the Erik Drift either in icebergs or in sea ice. The paper provides no information on the chronology other than to say it is discussed in a paper that is listed as "in press" but it is not in the reference list. It is also important, in my view, to state what has been used for the ocean reservoir correction and was an error attached to the value? This issue limits how well the chronology can be defined, hence the reliability of correlations with other records. It is a difficult issue that bedevils all of us (see ref. to Sjerup et al. 2010, their ref list). The authors note that Jennings et al (2014) were not able to identify a specific Icelandic tephra in the last 1 cal ka or so, hence it is difficult to constrain the possible $\Delta$R.

REPLY

We decided to publish the chronology in this article since it may be published before the one cited in the text (Kleiven et al in prep). We obtained a total of 12 accelerator mass spectrometry (AMS) 14C dates, based on the calcareous shells of the planktonic foraminifera Neogloboquadrina pachyderma (sinistral). The dates were analyzed on the Accelerator Mass Spectrometer at the Leibniz Labor für Altersbestimmung und Isotopenforschung in Kiel, Germany. Radiocarbon ages have been converted into calendar years using the CALIB (rev 5.0.1) software (Stuiver and Reimer, 1993) in conjunction with the Marine04 calibration dataset (Hughen et al., 2004). All dates were calibrated with a constant surface reservoir age of 400 years. The sample at 0 cm showed erroneous age because of severe addition of more than 100% modern carbon (pMC) and is assumed to be post-AD 1962 (relative to the increase in bomb radiocarbon levels in the North Atlantic region). The core was collected in 2006 and the Cesium spike in 210Pb in the upper 12 cm of the core sediments confirms post-AD 1964 age. A table with the uncorrected 14C ages and calibrated ages is provided in the revised version.

RC

I feel quite strongly that there needed to be more discussion on rationale for choosing the > 63 $\mu$m fraction as an IRD signal (Andrews, 2000). I think the only really unambiguous IRD grain-size signal are clasts > 2 mm (Grobe, 1987), although a solid case can be made for a $\geq$ 250 $\mu$m. When the fine sand and greater fractions are being identified, especially on a Drift, then I think an initial analysis should include the entire grain-size spectra (Prins et al., 2002) as this, typically, indicates IRD as a distinct hump at the coarse end of the grain-size spectra.

REPLY

In this case, we wanted to compare our IRD records with Gerad Bond's records and therefore we chose the 63-150 $\mu$m fraction, as he did for his publications. Bond's technique (Bond et al., 1997) was robustly tested using several multicores in the polar-subpolar region and it was compared to counts in the >150 $\mu$m fraction. We acknowledge that grains >250 $\mu$m are the best fraction to claim transport by icebergs and sea ice because wind and deep currents can be ruled out. Unfortunately, the study interval does not contain enough grains of this fraction to develop a sound analysis, not even a preliminary one to show trends in the IRD, we will probably need larger amounts of bulk sediment to perform a decent count of IRD >250 $\mu$m. Even though it has been suggested that within the 63-150 $\mu$m fraction some grains might be transported by other means (see discussion in (Andrews et al., 2014)), given the location of the study site (in the outer part of Eirik Drift) we think meltwater plumes are very unlikely and deep currents hardly transport sediments >63 $\mu$m, and therefore we can assume the 63-150 $\mu$m fraction we studied is mainly composed of IRD grains.

RC

I also note that there is no discussion on iceberg history (e.g. (Bigg, 1999; Bigg et al., 2014; Bigg and Wilton, 2014)) or on sea ice, especially the export of the "storis" (Schmith and Hanssen, 2003).

REPLY

This will be added to the discussion, thanks for the suggestion.

RC

Finally, the discussion of the provenance of the > 63 $\mu$m fraction might have usefully identified (on their Fig. 1?) the major tidewater ice streams/glaciers of SE/E/NE Greenland and have referenced the likely annual flux (km3/yr) versus that of sea ice, this would help in trying to establish provenance. For example, coal outcrops in the area of Nansen Fjord, East Greenland, and it has been recorded in sediments on the inner shelf (Jennings, person. Commun. 2010) but I am not sure if this was stated in any of her publications. The issue of the source(s) HSG is an important one given the attention it achieved through Gerard Bond's work. The most probable source is the Devonian outcrop ca 73°N, NE Greenland (Larsen et al., 2008) in the area of Kasjer Franz Joseph Fjord. Several cores were taken from this area during a Polarstern cruise (Evans et al., 2002; Hubberten and al., 1995; Stein, 2008) although evidence for significant IRD output over the last millennium is muted and the number of tidewater glaciers on the outcrop is limited.

REPLY

Figure 1 was modified to show the main tidewater ice streams we refer in the text: Helheim (H), Kangerdlugssuaq (K), Nansen (N), and Scoresby Sund (SS).

This discussion is very interesting, particularly regarding to the HSG sources, which is one of the main evidences here to show Arctic ice export. The relative abundance of coal is significantly low and we decided it was not solid enough to discuss possible

sources, plus there is not much work done with this type of rock, and therefore, we may be missing important sources. About the HSG sources, indeed in Alonso-Garcia et al. {Alonso-Garcia, 2013 #1646}, we showed the potential sources within Greenland, and we also suggested the potential input from Arctic sea ice transporting HSG from Northern Greenland and Canada as well as from the Svalbard-Franz Josef Land region.

We are extending the discussion in order to provide a more solid context for our hypothesis about the linkage between Arctic ice export and HSG deposition.

References

Andrews, J. T., Bigg, G. R., and Wilton, D. J.: Holocene ice-rafting and sediment transport from the glaciated margin of East Greenland (67–70°N) to the N Iceland shelves: detecting and modelling changing sediment sources, Quat. Sci. Rev., 91, 204-217, 2014. Bond, G., Showers, W., Cheseby, M., Lotti, R., Almasi, P., deMenocal, P., Priore, P., Cullen, H., Hajdas, I., and Bonani, G.: A Pervasive Millennial-Scale Cycle in North Atlantic Holocene and Glacial Climates, Science, 278, 1257-1266, 1997.

[Figure]

[Figure]

Figure 1. A) Location of multicore GS06-144-03 (red star) and other sites in the Northern North Atlantic whose records have been used to support the hypothesis proposed in this work. General North Atlantic circulation is shown according to Schmitz and McCartney (1993). The location of Norse settlements in Greenland is shaded and indicated with ES (Eastern settlement) and WS (Western settlement). The location of the main tidewater ice streams at present in SE Greenland is also depicted: Helheim (H), Kangerdlugssuaq (K), Nansen (N), and Scoresby Sund (SS). B) Temperature and salinity profiles of the first 1000 m at site GS06-144-03 obtained though Ocean Data View (http://odv.awi.de/en/home/) from the World Ocean Atlas 2013 (Locarnini et al., 2013; Zweng et al., 2013).

**Fig. 1.** Revised version of figure 1

---

## Author Comment (AC3) · 26 Oct 2016

Response to Reviewer#3 on "Freshening of the Labrador Sea as a trigger for Little Ice Age development" by Montserrat Alonso-Garcia et al.

We kindly thank Reviewer#3 for his very valuable comments. We really appreciate the feedback provided by this reviewer regarding to atmospheric processes, and ice sheet-ocean interactions.

In order to provide context to our replies, the referee's comments have been copied below preceded by "RC" and our replies are preceded by "REPLY". We agree with most of the suggestions to reformulate the text, so below we only included the discussion comments for which we can give an answer.

[Figure]

RC

Alonso-Garcia and coauthors present high-resolution sediment data from Eirik Drift off the southern coast of Greenland. They analyze the amount and mineralogy of icerafted debris (IRD), covering approximately the last 2000 years. The primary results are several episodes of increased deposition of IRD and an increase in hematite stained grains after about 1400 AD. This leads the authors to suggest that Arctic sea ice export strengthened and the Atlantic subpolar gyre (SPG) weakened. As a consequence, less Atlantic waters were present in Greenland fjords, changing the stability of tidewater glaciers, their calving rates and thus the amount of IRD transported to the core site. The original data is compared with several older records from the same region in the subpolar North Atlantic. I think this study contains very valuable new data and that it is a good addition to the existing literature (Copard et al., 2012; Moffa-Sanchez et al. 2014a, Moreno-Chamarro et al., 2016). Text and figures are mostly clear. Previous works are referenced adequately. I do not fully agree with the interpretation of the data, which is rather speculative on several occasions and likely incorrect in some aspects. I will summarize my criticism in 4 major points: 1) Several of the claims made in the discussion are not well supported by the data. In my opinion, the sediment record does not warrant statements about the stability of the calving front of Greenland glaciers. Fjord environments are extremely complex and heterogeneous in their dynamics so that no robust conclusions can be drawn from a single sediment core several hundreds or thousands of miles away. Ice-rafting occurs both during 'cold' and 'warm' episodes, for which two different and only weakly supported explanations are given. Furthermore, whether the MCA and the LIA periods really had a temperature signal in the relevant regions has not been shown in the manuscript. I think they are better characterized as high and low sea ice periods.

REPLY

We agree that finding an explanation for ice-rafting during both "warm" and "cold"periods is challenging, particularly if you see higher IRD deposition during the

MCA. Here the reviewer suggests variations in sea ice as a better explanation for the IRD input. I guess the reviewer means higher/lower sea ice export from the Arctic rather than in situ sea ice conditions, since our site is quite far from the continent. However, we find it difficult to argue in favor high sea ice export during the MCA based on the available proxy records. Instead, we find evidences for increases in sea ice export after ~1300 yr AD (e.g. (Andrews et al., 2009; Massé et al., 2008), in agreement with the increase in our HSG record. Moreover, our hypothesis of intense calving in SE Greenland is based on records from the SE Greenland coast (Jennings and Weiner, 1996; Miettinen et al., 2015), which show rather high SST, low sea ice coverage and Atlantic water presence in the fjord of this region from 1000 to 1200 yr AD. This interval coincides with the interval of high IRD deposition in our record, therefore, we argued that intense calving occurred in SE Greenland fjords during the late MCA due to warm temperatures and the presence of Atlantic waters.

RC

2) Throughout the paper, references are made to an outdated reconstruction of the North Atlantic Oscillation (NAO) (Trouet et al., 2009). It has been shown that the method of this reconstruction is flawed and that its results are not trustworthy (Lehner et al., 2012). A new and more advanced method shows very different results (Ortega et al., 2015). It is clear that "some" change in the atmospheric circulation took place at 1400 AD that led to changes in the proxy records (Trouet et al., 2009; Olsen et al., 2012), but that change is probably unrelated with the NAO. I really do not see the necessity to relate the original data of this manuscript with the NAO and thus recommend to remove this link altogether.

REPLY

This was also commented by Reviewer#1 and we are eliminating all the discussion related to NAO, as well as the NAO references in figures 3 and 5 (see response to Reviewer#1).

RC

3) The delay between the onset of the stronger freshwater forcing at _1000 AD and the weakening of the SPG _1250 AD, as well as the further lag of 200 years before the cooling of Atlantic waters in the fjords (page 12) are not reasonable and not well supported by evidence. Present-day observations and modeling show that a slowdown of the SPG after weakening the convection in the Labrador Sea takes place within a single season or at most some few years. There is no physical reason to assume a delay of multiple centuries. I believe the same is true for the cooling of Greenland fjords, although I do not know that for certain. I suggest the authors estimate the energy balance and fluxes to support their claim or remove this part. On a more general note, the authors seem to expect a strict determinism underlying their data, which is probably not correct. In the highly variable North Atlantic and due to the strong positive feedbacks associated with the SPG, abrupt changes can happen suddenly and in response to rather minor forcing pulses (Moreno-Chamarro et al., 2016). Not every wiggle will have an easily identifiable cause.

REPLY

We agree with the reviewer that not finding an explanation for every is difficult. This part of the discussion is being rephrased eliminating the interpretations about lags in the forcings.

RC

4) While a weakening of the SPG at about 1400 AD is consistent with findings by Copard et al. (2012), Moffa-Sanchez et al. (2014a) report more frequent changes in the strength of the gyre. How can these views be reconciled? This should be included in the discussion.

REPLY

The subpolar gyre entered in a weaker mode at ∼1300 AD, according to Moffa-

Sanchez et al. (2014a), and after that they registered some SST and salinity oscillations in the record south of Iceland, which indicate oscillations in the subpolar gyre. However, their G. bulloides record from the Labrador Sea (see supplementary data from the same article, Moffa-Sanchez et al. (2014b), and fig. 4 of this manuscript) indicates the Labrador Sea remained rather fresh and cold after 1300 AD. Indeed the Labrador Sea started to get colder at ∼1200 AD. This discussion will be added to the manuscript

RC

minor comments: l 95: Say when this shift occurred.

REPLY

At ∼1350 AD. This will be added in the final version.

RC

l 137: Add a reference.

REPLY

(Born and Stocker, 2014)

RC

l 146: How can robustness be claimed when the key publication by one of the coauthors has not even been submitted?

REPLY

The age model for this record has been included and will be published in this paper (see response to reviewer#2).

RC

l 215: This argument is not convincing. If the westerly winds prevented dust from

being transported upwind to the sedimentation site, they would also have avoided dust from reaching Greenland glaciers as well. However, the climatological wind direction is irrelevant here, because volcanic eruptions are usually short-lived events and dust is transported in the direction of wind at that time. It can not be ruled out that a volcano erupted during a week of predominantly easterly winds, even though that may be the less common situation.

REPLY

With this statement, we just wanted to say that atmospheric conditions do not favor the deposition the volcanic shards in the study area and maybe this is why we can't find any specific layer of volcanic ash. This is going to be rephrased in order to clarify our statement.

RC

l 246: This argument shows that while the small recent variations in Arctic sea ice export correlated with the NAO, large anomalies in export probably have a different cause, like the GSA. Maybe sea ice during the LIA was thicker in the Arctic and so more ice (volume) was exported without a faster flow? This view would be consistent with findings from Miller et al. (2012) and Lehner et al. (2013). Also, I am unsure whether the manuscript discusses the impact of freshwater forcing from sea ice.

REPLY

NAO discussion has been removed.

References

Andrews, J. T., Belt, S. T., Olafsdottir, S., Massé, G., and Vare, L. L.: Sea ice and marine climate variability for NW Iceland/Denmark Strait over the last 2000 cal. yr BP, The Holocene, 19, 775-784, 2009. Born, A. and Stocker, T. F.: Two Stable Equilibria of the Atlantic Subpolar Gyre, J. Phys. Oceanogr., 44, 246-264, 2014. Jennings, A. E. and Weiner, N. J.: Environmental change in eastern Greenland during

the last 1300 years: evidence from foraminifera and lithofacies in Nansen Fjord, 68°N, The Holocene, 6, 179-191, 1996. Massé, G., Rowland, S. J., Sicre, M.-A., Jacob, J., Jansen, E., and Belt, S. T.: Abrupt climate changes for Iceland during the last millennium: Evidence from high resolution sea ice reconstructions, Earth Planet. Sci. Lett., 269, 565-569, 2008. Miettinen, A., Divine, D. V., Husum, K., Koç, N., and Jennings, A.: Exceptional ocean surface conditions on the SE Greenland shelf during the Medieval Climate Anomaly, Paleoceanography, 30, 1657-1674, 2015. Moffa-Sanchez, P., Born, A., Hall, I. R., Thornalley, D. J. R., and Barker, S.: Solar forcing of North Atlantic surface temperature and salinity over the past millennium, Nature Geosci, 7, 275-278, 2014a. Moffa-Sanchez, P., Hall, I. R., Barker, S., Thornalley, D. J. R., and Yashayaev, I.: Surface changes in the eastern Labrador Sea around the onset of the Little Ice Age, Paleoceanography, 29, 2013PA002523, 2014b.